# Unveiling the Role of Data Uncertainty in Tabular Deep Learning

## Abstract

Recent advancements in tabular deep learning have demonstrated exceptional practical performance, yet the field often lacks a clear understanding of why these techniques actually succeed. To address this gap, our paper highlights the importance of the concept of data uncertainty for explaining the effectiveness of recent tabular DL methods. In particular, we reveal that the success of many beneficial design choices in tabular DL, such as numerical feature embeddings, retrieval-augmented models, and advanced ensembling strategies, can be partially attributed to their implicit mechanisms for performing well under high data uncertainty. By dissecting these mechanisms, we provide a unifying understanding of recent performance improvements. Furthermore, the insights derived from this data-uncertainty perspective directly allowed us to develop more effective numerical feature embeddings as an immediate practical outcome of our analysis. Overall, our work paves the way toward a foundational understanding of the benefits introduced by modern tabular methods that results in the concrete advancements of existing techniques and outlines future research directions for tabular DL.

## 1 Introduction

Deep learning for tabular data is experiencing rapid progress, with new models and training approaches constantly improving performance across a wide range of tasks (Gorishniy et al., 2022; Hollmann et al., 2023; Gorishniy et al., 2024; Ye et al., 2024; Holzmüller et al., 2024; Gorishniy et al., 2025; Hollmann et al., 2025). However, a clear understanding of why the proposed techniques are effective mostly lags behind their empirical success. New methods are often introduced based primarily on empirical observations or by repurposing ideas from other domains, often without a thorough investigation of their effectiveness within the tabular data context. This gap between performance and understanding can hinder future progress, potentially leading to a more trial-and-error research style.

In this paper, we address this gap by leveraging the concept of *data (aleatoric) uncertainty* (Gal et al., 2016) as an informative tool for analyzing and understanding the effectiveness of recent tabular DL methods. Our rough motivation to employ data uncertainty originates from the intuition that, compared to computer vision or NLP, tabular problems often possess unobserved confounding variables and inherent noise in target labels that make it challenging for models — and even human experts — to achieve perfect predictive accuracy due to irreducible noise or ambiguity. Furthermore, much indirect evidence also indicates that high data uncertainty is a significant factor in tabular learning. For instance, the critical reliance of tabular DL performance on early stopping hints at the presence of significant noise in the labels (Baek et al., 2024). While any single point above does not strictly necessitate that data uncertainty is an important constituent of a typical tabular problem, accumulated evidence from them convinced us to take a closer look at this concept. Interestingly, in the preliminary experiments, we compared a simple MLP model to the leading GBDT implementation in terms of performance on datapoints with different data uncertainty. For several datasets, Figure 1 shows that GBDTs perform much better specifically in the high data uncertainty niche, which indicates that the "DL vs GBDT" battle probably unfolds mostly there.

In light of this, we systematically investigate several influential findings from the tabular DL literature — specifically, numerical feature embeddings (Gorishniy et al., 2022), retrieval-augmented models (Ye et al., 2024), and advanced ensembling strategies (Gorishniy et al., 2025) — in terms of their ability to handle data uncertainty. Our investigation reveals a compelling and consistent pattern: the

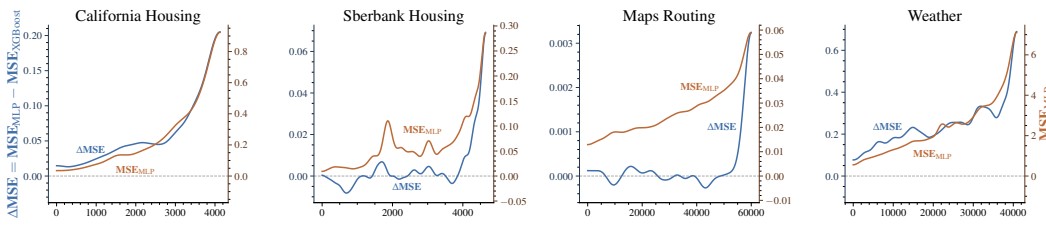

Figure 1: The performance differences measured by MSE between MLP and XGBoost are shown in blue, and absolute values of MSE for MLP are in brown. The figure has two separate vertical axes: the left ones correspond to $\Delta$MSE, and the right ones correspond to absolute MSE. Metrics are reported for each test example, sorted left-to-right by data uncertainty. On four datasets, especially on Sberbank Housing and Maps Routing, MLP has significantly worse performance compared to XGBoost on high data uncertainty points. On Weather, this phenomenon is more subtle, but in the region of high uncertainty, the MSE difference still grows faster than MSE itself.

techniques often appear to be disproportionately more beneficial in highly uncertain regions of the data. To obtain a clearer understanding, for each technique we dissect the specific mechanisms that enable better learning in this challenging niche.

As a direct practical outcome of our analysis, we develop a novel numerical feature embedding scheme that is more effective than existing alternatives.

Overall, the main contributions of our paper are:

- We introduce data uncertainty as an important tool for analyzing the performance of various tabular learning methods and demonstrate that it provides an informative viewpoint on the methods' advantages.
- We provide a mathematical intuition behind the effectiveness of several recent tabular DL techniques in the presense of high data uncertainty, including numerical feature embeddings, retrieval-augmented models, and parameter-efficient ensembling.
- We leverage the insights from our uncertainty-driven analysis to design a new, more effective numerical feature embedding scheme.

## 2 RELATED WORK

**Tabular Learning.** The field of tabular data modeling has seen a significant transformation in recent years, with deep learning approaches increasingly challenging the long-held supremacy of traditional "shallow" decision tree-based ensembles, such as Gradient Boosting Decision Trees (GBDTs). In particular, the most recent DL models (Gorishniy et al., 2024; Holzmüller et al., 2024; Ye et al., 2024; Gorishniy et al., 2025; Hollmann et al., 2025) have compellingly demonstrated performance on par with, or even higher, than that of leading GBDT implementations like CatBoost (Prokhorenkova et al., 2018), LightGBM (Ke et al., 2017), and XGBoost (Chen & Guestrin, 2016). This practical success of tabular DL is a consequence of recent research efforts on novel architectures (Gorishniy et al., 2021; Somepalli et al., 2021; Holzmüller et al., 2024; Gorishniy et al., 2022; Ye et al., 2024; Gorishniy et al., 2024; 2025), specialized regularizations and learning protocols (Bahri et al., 2021; Rubachev et al., 2022; Jeffares et al., 2023; Thimonier et al., 2024). However, despite this plethora of empirically successful techniques, an understanding of the core principles behind their effectiveness often remains unclear, highlighting a need for deeper analysis of their underlying mechanisms and comparative strengths.

**Analysis in Tabular DL.** Several recent papers also contribute to a deeper understanding of the strengths and limitations of tabular DL methods. Grinsztajn et al. (2022) identifies the reasons why tabular DL can be inferior to GBDT methods on certain datasets and formulates the desiderata for tabular DL methods, in particular, the ability to model irregular target dependencies. McElfresh et al. (2023) analyses the properties of tabular datasets that can be indicative of whether DL or GBDT models should be preferred. Rubachev et al. (2025) re-evaluates a large number of recent design choices in tabular DL on a more realistic benchmark and reveals that many of them are not robust to

a temporal train–test shift. In contrast to these dataset-centric or technique-centric investigations, our paper proposes a more fine-grained sample-wise analysis, examining performance at the individual datapoint resolution.

**Uncertainty Estimation.** Uncertainty is a core concept in machine learning, reflecting potential inaccuracies in model predictions (Gal et al., 2016). In more detail, the total uncertainty can be decomposed into **knowledge** (or **epistemic**) and **data** (or **aleatoric**) uncertainty (Depeweg et al., 2017). Our work concentrates on data (aleatoric) uncertainty, which is inherent to the data itself and is independent of the specific model used — typically arising from noise or randomness in the underlying process being modeled (e.g., noisy features or targets).

## 3 PRELIMINARIES

In this section, we outline the core concepts and notation required to analyze how different tabular deep learning approaches interact with data uncertainty.

The concept of data uncertainty arises when the target dependency is not deterministic, and instead the relationship between features $x$ and target variable $y$ is set by a non-degenerate conditional distribution $p(y \mid x)$. This can occur for several reasons, for example, in some cases, the label-collection procedure can be noisy due to mistakes of human annotators or due to imperfect sensors. In other cases, features in the dataset do not strictly determine the target variable; for example, it is impossible to know an exact temperature in a given city by its longitude and latitude alone. In these cases, the labels provided in a given dataset are samples from this conditional distribution $p(y \mid x)$.

The distribution $p(y \mid x)$ may have different characteristics for different $x$. For example, measurements of a target quantity could be exact for one set of $x$ and only an approximation for another. The concept of **data uncertainty** is used to measure how "spread out" the distributions $p(y \mid x)$ are for different samples of the data.

Informally, **data uncertainty** quantifies the inherent noise, particularly label noise, present in the data. While the loss in performance that comes from label noise in the test data is irreducible, it has been shown (Tanaka et al., 2018) that when training on noisy labels, the performance of a model decreases even when measured on a clean test set. Different approaches and models can be influenced by this decrease to a different degree, which gives rise to a phenomenon we study in this paper.

Our analysis focuses on regression tasks, in which data uncertainty for sample $i$ is defined as the variance of the distribution $p(y_i \mid x_i)$. In section 5, section 6, and section 7, we show how increases in performance for recent methods in tabular deep learning are disproportionately larger on samples where data uncertainty is higher.

### 3.1 DATA UNCERTAINTY ESTIMATION

A large part of our analysis relies on the empirical estimates of data uncertainty. When estimating data uncertainty, we assume the following probabilistic model:

$$y_i = f(x_i) + e^{g(x_i)} \cdot \mathcal{N}(0, 1) \tag{1}$$

Here $(x_i, y_i)$ represents a datapoint, where $x_i$ is the feature set and $y_i$ is the target variable. The deterministic function $f(x_i)$ is the noiseless part of the target dependency, while $e^{2 \cdot g(x_i)}$ represents the data uncertainty[1]. To obtain our estimates of data uncertainty, we train a machine learning model to predict both $(f(x_i), g(x_i))$ for any given datapoint $x_i$. The model is trained by maximizing the likelihood of the observed target values under a distribution $\mathcal{N}(f(x_i), e^{2 \cdot g(x_i)})$. This estimation approach aligns with the recent works (Duan et al., 2020; Malinin et al., 2021). In our experiments, we use CatBoost models (Prokhorenkova et al., 2018) to estimate the data uncertainty values, though any suitable deep learning model could be employed as well.

Crucially, we find that the predicted data uncertainty values $e^{2 \cdot g(x_i)}$ are stable regardless of the specific estimator model used. For instance, see Figure 2a, which demonstrates the strong correlations

---

[1]This parameterization ensures that data uncertainty values are always positive.

of uncertainty estimates produced by CatBoost and MLP on the California Housing, Maps Routing and Weather datasets. Moreover, the effects we analyze in section 5, section 6 and section 7 stay the same regardless of which uncertainty estimation method we use, as shown in Appendix I.

To confirm the reliability of our empirical data uncertainty estimates even further, we generate a synthetic dataset where both functions $f(\cdot)$ and $g(\cdot)$ from Equation 1 are randomly initialized MLPs. Figure 2b shows that the uncertainty values predicted by CatBoost closely match the ground-truth uncertainty values on this synthetic data. Moreover, we compare different tabular DL methods on datasets described in subsection 4.2 and subsection 4.1 and demonstrate that an increase in performance coincides with high data uncertainty even when the true values of data uncertainty are known and no estimation is used in the analysis.

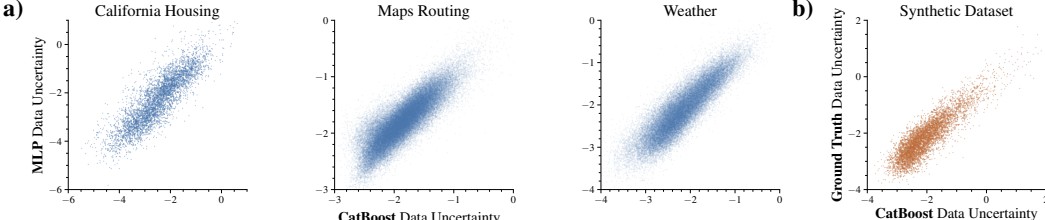

Figure 2: **a)** The scatterplots show a high correlation between the log-transformed data uncertainty estimates from CatBoost and an MLP. **b)** The scatterplot shows the relationship between CatBoost's log-estimated data uncertainty and the logarithms of the true data uncertainty values used for target sampling in the controlled synthetic experiment.

To analyze how various tabular DL methods perform on datapoints with different levels of data uncertainty, we draw *uncertainty plots* as follows. First, we obtain an estimate of data uncertainty for each test sample in the dataset as described above. Then, test datapoints are sorted by increasing values of their estimated data uncertainty (or true data uncertainty for synthetic data). For each datapoint, we then compute the performance (in terms of Mean Squared Error) of the simple MLP model and the method whose behavior we aim to analyze. Then, for visualization purposes, we additionally smooth the resulting curve (see the details in Appendix B). As an illustrative example, Figure 1 shows the uncertainty plot that compares MLP and XGBoost on several datasets and reveals that MLP has much worse performance on datapoints of high data uncertainty.

## 4 SYNTHETIC DATA

For real datasets, it is not entirely clear whether our estimation of data uncertainty reliably represents the true data uncertainty. For this reason, we first use synthetic datasets to demonstrate how the performance of several tabular DL methods changes at different levels of data uncertainty.

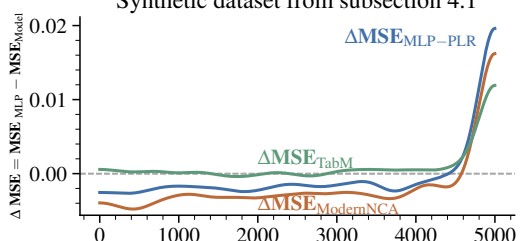

Figure 3: This figure reports the performance differences on the synthetic dataset described in subsection 4.1 The test examples are sorted in order of increasing data uncertainty. We report differences in MSE between MLP and each of MLP-PLR, ModernNCA and TabM. Differences are reported for each test example, sorted from left to right by increasing true data uncertainty.

### 4.1 SYNTHETIC DATASET WITH DATA UNCERTAINTY PARAMETERIZED BY AN MLP

The first dataset we use is obtained in the following way. First, we perform i.i.d. sampling from the 20-dimensional standard Gaussian distribution to obtain the feature vectors $x_i$. Then, we produce the target variables as $f(x_i) + e^{g(x_i)} \cdot \mathcal{N}(0, 1)$, where both $f(\cdot)$ and $g(\cdot)$ are parameterized as randomly initialized MLPs. As a result of this procedure, we obtain a dataset for which data uncertainty exists and we know its true values for each sample. For full details regarding the creation of this dataset, see Appendix F. First, using this dataset, in Figure 2b we show that when estimating data uncertainty with CatBoost, the obtained estimates are higly correlated

with the true values. Moreover, in Figure 3, we demonstrate that when using true data uncertainty, MLP-PLR, ModernNCA and TabM still show increased performance compared to a simple MLP on the high data uncertainty region of the data.

## 4.2 SAW-LIKE 2D SYNTHETIC DATASET

In the previous subsection, we demonstrated that the true values of data uncertainty are correlated with increases in the performance of numerical embeddings, ModernNCA, and TabM. As an additional illustration, in this section we visualize this behavior for the $2D$ dataset presented in Figure 4. In this dataset, the noiseless target dependency is relatively simple and is shown in color on the left part of Figure 4. Data uncertainty is obtained by adding noise with variance increasing with the feature $x_2$, which corresponds to the vertical direction in the plot. The standard deviation of the noise distribution is shown on the $x$-axis on the right side of Figure 4. For specifics on dataset generation, see Appendix D.

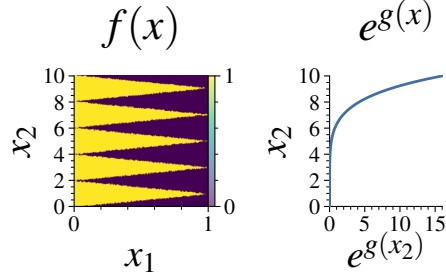

Figure 4: The plots show visualization of the synthetic data generation process used in subsection 4.2. The left plot shows the function $f(x)$ in color, while the right plot shows $e^{g(x)}$ on the $x$-axis. Targets $y$ are generated as $y = f(x) + e^{g(x)} \cdot \mathcal{N}(0,1)$.

Figure 5 demonstrates the predictions of the simple MLP model, as well as its following modifications: 1) Deep Ensembles — the average prediction of five independently trained MLP models. 2) ModernNCA (Ye et al., 2024) — a retrieval-based model that consists of an MLP backbone followed by a kNN-like layer, producing a prediction by aggregating targets of training datapoints. 3) TabM (Gorishniy et al., 2025) — a recent model that imitates an ensemble of MLPs in a parameter-efficient manner. 4) MLP-LRLR — an MLP model augmented with learned numerical feature embeddings, proposed in Gorishniy et al. (2022). We provide the exact descriptions of the hyperparameter tuning for each model in Appendix E.

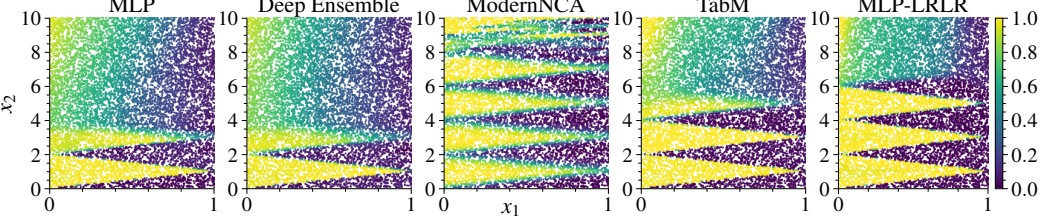

Figure 5: The plots show predictions of different models on the synthetic dataset described in subsection 4.2. Predicted values are shown in color, and data uncertainty increases bottom-to-top.

From Figure 5, we see that there are almost no differences in performance between Deep Ensembles and MLP on this dataset. Meanwhile, MLP-LRLR, ModernNCA and TabM show substantially improved performance in the regions of high data uncertainty – that is, the inner logic of these techniques somehow allows them to manage uncertain regions of the dataset better. Further in the paper, section 5, section 6, and section 7 analyze the mechanisms behind each model in more detail.

## 4.3 SYNTHETIC VERSION OF THE CIFAR-10 DATASET WITH INCREASED DATA UNCERTAINTY

In this section, we construct another synthetic dataset based on the established CIFAR-10 dataset (Krizhevsky et al., 2009). To do this, we use the intuition that data uncertainty may increase when some features relevant to the target variable are omitted from the dataset. Specifically, we take a standard version of CIFAR-10, which has $32 \times 32 \times 3 = 3072$ features, and then randomly choose 50 features to keep, dropping all the other features from each sample in the dataset. As a result of this operation, we obtain a dataset we refer to as Uncertain CIFAR-10.

We then conduct experiments on both the pruned and the original versions of CIFAR-10, using standard tabular DL methods. As can be seen from Table 1, on the original version of the dataset, MLP outperforms both other single model methods (MLP-LRLR and Modern-NCA), and in ensembling methods there is no difference between standard Deep Ensemble and TabM. However, on the Uncertain CIFAR-10, TabM outperforms Deep Ensemble, while MLP-LRLR and ModernNCA outperform MLP. This experiment provides additional evidence of a relationship between data uncertainty and increases in performance for different tabular methods.

Table 1: Accuracy of standard tabular DL methods on two versions of CIFAR-10. In each category (Single Models and Ensembles), the best method and methods with statistically insignificant differences from it are emphasized in bold.

| | Accuracy ↑ | |
|---|---|---|
| Method | CIFAR-10 | Uncertain CIFAR-10 |
| **Single Model Methods** | | |
| MLP | **58.31±0.27** | 47.54±0.17 |
| MLP-LRLR | 57.19±0.47 | **48.73±0.23** |
| ModernNCA | 57.48±0.22 | **48.45±0.30** |
| **Emsembling Methods** | | |
| Deep Ensemble | **60.92±0.22** | 49.41±0.20 |
| TabM | **61.01±0.25** | **50.45±0.19** |

## 5 EMBEDDINGS FOR NUMERICAL FEATURES

Recent work by Gorishniy et al. (2022) demonstrated that, for tabular DL, embedding scalar numerical features into a high-dimensional space before they are fed into the main backbone is beneficial. While numerical embeddings have become an established design choice in the field, the reasons for their effectiveness are not entirely clear. In our paper, we reveal that an important property of numerical embeddings is their ability to handle high data uncertainty regions of data.

Figure 6 demonstrates uncertainty plots for MLP vs MLP-PLR, where "-PLR" denotes periodic activations followed by a linear layer with a ReLU nonlinearity as described in Gorishniy et al. (2022). Here we report the analysis on four datasets[2]: 1) California Housing[3], an established regression dataset, where numerical embeddings were shown to be beneficial (Gorishniy et al., 2022). 2) Sberbank Housing (Rubachev et al., 2025), a recently introduced dataset with a temporal distribution shift between its training and test subsets. 3) Maps Routing (Rubachev et al., 2025), a large-scale dataset needed to demonstrate that our analysis is valid for larger problems as well. 4) Delivery ETA (Rubachev et al., 2025), a dataset where MLP-PLR underperforms MLP. Despite this, the model with embeddings still slightly outperforms the MLP baseline in the high data uncertainty region, while losing more in the lower data uncertainty niche.

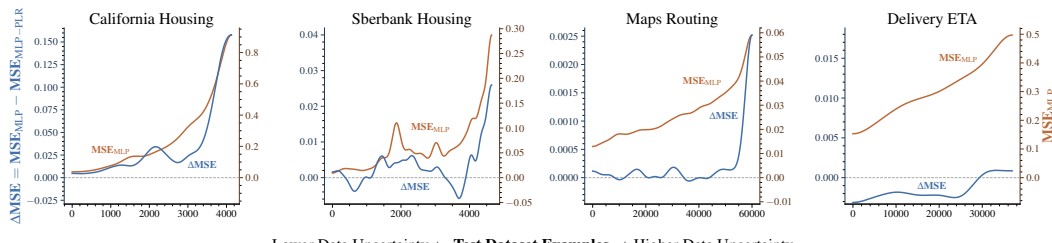

Figure 6: Uncertainty plots report the differences in MSE between MLP and MLP-PLR. The increase in performance is the most significant in the high data uncertainty niche.

The main observation from Figure 6 is that numerical "-PLR" embeddings are disproportionately more beneficial for high data uncertainty samples. Specifically, the difference in MSE grows faster with data uncertainty than the value of MSE for the baseline MLP, which shows that numerical embeddings do not just proportionally improve performance for all levels of uncertainty, but do so more in high data uncertainty region.

---

[2]Similar plots on other established datasets are reported in Appendix A.
[3]https://www.dcc.fc.up.pt/~ltorgo/Regression/cal_housing.html

## 5.1 WHY DO NUMERICAL EMBEDDINGS OUTPERFORM IN THE HIGH DATA UNCERTAINTY NICHE?

When using numerical embeddings, one maps the original datapoints into a higher-dimensional space that effectively becomes a new input space for the main backbone. We claim that properly designed numerical embeddings correspond to a space with higher local target consistency, i.e., the neighboring points in this space are more likely to have similar targets. We demonstrate that behavior on several datasets. Figure 7 reports the squared difference between the target of a test datapoint and the target of its $k$-th closest neighbor from the training set (averaged over all test subset) for MLP and MLP-PLR models. We sort the neighbors by the Euclidean distance in the latent space obtained after the first block of the model, to show the distance in the space that has already incorporated the signal from all features and still comes relatively early in the model.

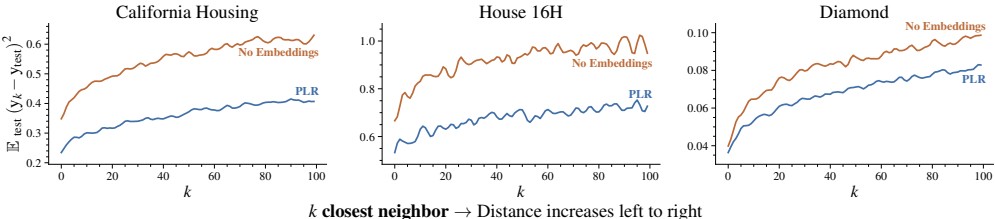

Figure 7: Squared difference between a test sample's target and the target of its $k$-th closest neighbor, averaged over the test set. The difference is smaller when using PLR embeddings.

While numerical embeddings improve the local target consistency in the neighborhoods of all datapoints, this leads to disproportionate performance improvement on datapoints that have higher data uncertainty, as shown in Figure 6. We explain this effect by the fact that the model's predictions on high data uncertainty points rely on the quality of their neighborhoods more heavily. Figure 5 confirms our intuition on the synthetic task. For low data uncertainty regions (bottom), a simple MLP is able to model sharp separating planes between the predictions "0" and "1". As data uncertainty increases, the MLP predictions become smoother, until they become so smooth that the original saw-like target structure becomes unnoticeable. This demonstrates that when MLP makes a prediction on a high data uncertainty datapoint, it is more dependent on the datapoint's neighborhood, effectively performing more "conservative" local target smoothing. As an additional illustration, Figure 5 shows that the better representations provided by the learned LRLR embeddings lead to better predictions even at the levels where the original MLP predictions fail.

### 5.2 A MORE EFFECTIVE EMBEDDING SCHEME

While numerical embeddings proposed in Gorishniy et al. (2022) substantially improve the quality of neighborhoods needed to handle high data uncertainty, they do not explicitly aim for this neighborhood-improving behavior. In contrast, we propose a new numerical embedding scheme that is learned by explicitly maximizing local target consistency. In more detail, we train an embedder module that produces a high-dimensional representation of a given object using the triplet loss (Schroff et al., 2015) to bring embeddings of objects with similar targets closer in the latent space. The specifics of the training procedure and hyperparameter spaces are described in Appendix G.

Table 2: Comparison of MLP, MLP-LRLR and MLP-LRLR trained with triplet loss proposed here. We report average ranks and median improvement over the MLP baseline. For full dataset-wise results and more details, see Appendix G.

| Model | $\Delta$ MLP (%) | Avg. Rank |
|---|---|---|
| MLP | 0.00 | 2.28 $_{(\pm 0.75)}$ |
| MLP-LRLR | 1.05 | 1.50 $_{(\pm 0.51)}$ |
| MLP-LRLR$_{\text{triplet}}$ | **1.43** | **1.06** $_{(\pm 0.24)}$ |

Table 2 demonstrates how our embedding scheme improves performance in comparison with LRLR embeddings from Gorishniy et al. (2022) trained from scratch. Information regarding statistical significance, along with the full results for each dataset, is provided in Appendix G.

## 6 MODERNNCA AND HIGH DATA UNCERTAINTY

To investigate the interplay between retrieval modules and data uncertainty, we focus on the ModernNCA model (Ye et al., 2024), which is a recent model that consists of an MLP-like backbone followed by a kNN-like prediction head. Figure 8 shows that compared to MLP, ModernNCA also improves the performance on higher data uncertainty points, while the performance on lower data uncertainty points sometimes slightly decreases. We explain this by first noticing that averaging over a large number of neighbors inherent to retrieval-augmented models leads to less overfitting on the high data uncertainty regions when compared to MLP.

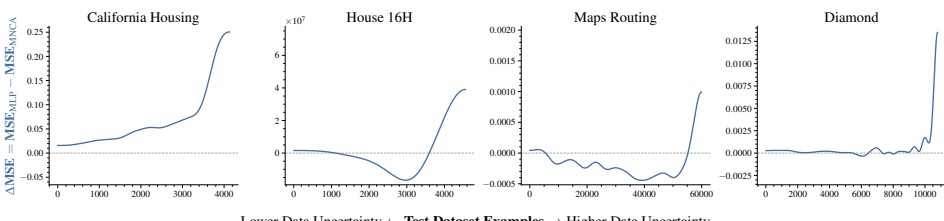

Figure 8: Uncertainty plots, built in the same way as in Figure 1, but showing difference in MSE between MLP and ModernNCA. On House and Maps-Routing datasets, ModernNCA is inferior to MLP in the lower/medium data uncertainty niches, while on the other two datasets it shows improvement in all niches of uncertainty.

To demonstrate this, in Figure 9 we report the sample-wise MSE loss of ModernNCA computed on training datapoints, where training datapoints are sorted in order of the increasing data uncertainty. Figure 9 shows that compared to MLP, ModernNCA severely underfits the data on high data uncertainty zone. On the one hand, this underfitting effect provides robustness to the noise in the train targets for the high data uncertainty niche, since ModernNCA memorizes the noise in the data to a lesser degree. On the other hand, on some datasets (e.g. Maps Routing, House), this excessive underfit can be harmful to performance in the lower and medium data uncertainty niches.

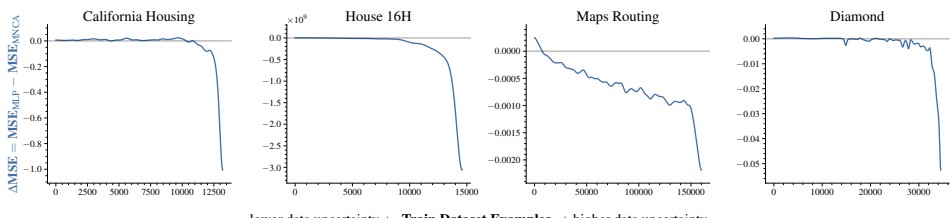

Figure 9: Difference in MSE performance of MLP and ModernNCA on train samples, with samples sorted by increasing uncertainty. ModernNCA's higher loss demonstrates that it often produces overly smooth predictions, underfitting the data, for example in Figure 5.

## 7 ENSEMBLING AND HIGH DATA UNCERTAINTY

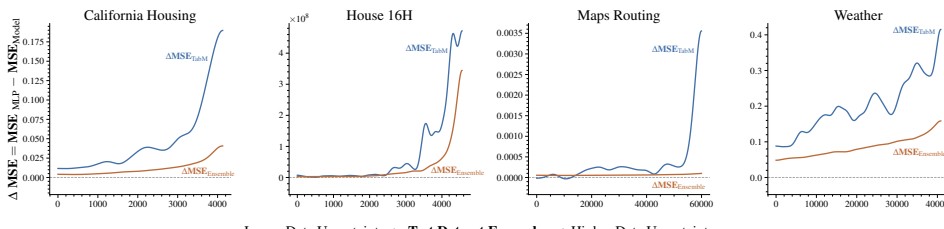

Figure 10: The plots show sample-wise differences in MSE between MLP and TabM in blue and sample-wise differences in MSE between MLP and a Deep Ensemble of MLPs in brown, with samples sorted by increasing data uncertainty. TabM shows substantially higher performance, especially on high data uncertainty regions.

The recent TabM model (Gorishniy et al., 2025) also demonstrates a more significant improvement on higher data uncertainty datapoints, as shown in Figure 10. Interestingly, while TabM was described

as an efficient ensemble, a traditional deep ensemble demonstrates inferior performance, which is especially noticeable in high data uncertainty regions. Another illustration of the differences between TabM and a deep ensemble is shown by our synthetic example described in subsection 4.2. While deep ensemble shows almost no difference in performance compared to a single MLP, TabM learns a function that much better represents the underlying noiseless target distribution.

In this section, we explain the reason behind the robustness of TabM to training on noisy targets compared to a deep ensemble. Below, we demonstrate that this stems from the fact that the averaging of the gradients for shared parameters of TabM effectively reduces the noise in the gradients from individual branches. For the synthetic dataset described in subsection 4.2, we know both the clean labels and the added noise, so we can decompose the gradient from the MSE loss into clean and noisy components. Then, we collect the clean and noisy components of the gradient for optimally tuned and trained TabM and MLP models. For TabM, we collect gradients across different branches as well as the averaged ones. For the specific details regarding gradient decomposition and collection, see Appendix H.

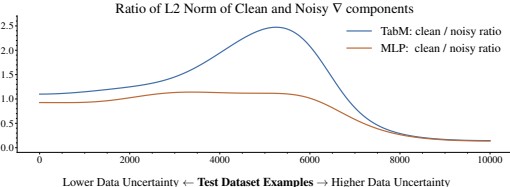
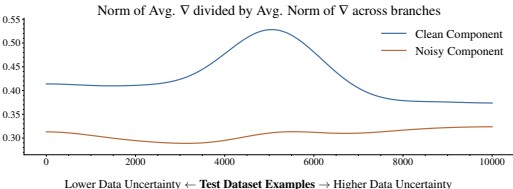

Figure 11: The plot shows ratios of the norms of the clean and noisy components of the gradient for TabM and MLP on the synthetic dataset described in subsection 4.2.

Figure 12: The plot shows the ratio of the average norm of a per-branch gradient to a norm of the gradient averaged between branches, taken separately for the clean and noisy components.

For robust training on noisy data, it is beneficial to minimize the noisy component of the gradients to make the overall gradients as close to their noiseless components as possible. In Figure 11, we show that the ratio of the clean to the noisy component of the gradient is much higher for TabM than for MLP, and this difference grows with data uncertainty until noise in the labels becomes large enough to overwhelm clean gradient component entirely for both models. This indicates that the noisy gradients affect TabM's learning process to a lesser extent. We explain this difference by averaging of the gradients obtained from different branches in TabM. To confirm this explanation, in Figure 12, we compute the ratio of the norm of the averaged gradient to the average norm of the gradients of the individual branches (for both the noisy and clean gradient components). As shown in the figure, the averaging reduces the clean component of the gradient less, and, as can be seen from Figure 5, this difference is highest in the region of data uncertainty where TabM outperforms Deep Ensemble the most. This behavior makes TabM more reluctant to learn noisy parts of the target dependency, which leads to better performance in the high data uncertainty region.

## 8 CONCLUSION

In this paper, we propose an uncertainty-centric lens to investigate tabular DL approaches. We analyse several recent tabular techniques and show that they are disproportionally more beneficial in regions of high data uncertainty. We provide explanations for these phenomena and, motivated by this analysis, we propose an improved numerical feature embeddings scheme that outperforms existing alternatives. Looking ahead, an interesting avenue for future research involves extending this uncertainty-driven analysis to the growing field of tabular foundational models (Hollmann et al., 2025).

## 9 LIMITATIONS

Several limitations should be acknowledged in this work. Our investigation was primarily focused on regression tasks; future research could extend this analysis to include classification problems. Furthermore, to maintain a specific focus on tabular DL approaches, we deliberately excluded the examination of general-purpose deep learning mechanisms (e.g., dropout, weight decay, robust optimizers). Lastly, although knowledge uncertainty presents an important and potentially informative viewpoint, its detailed exploration was not a central objective of the current study.

## REPRODUCIBILITY STATEMENT

To ensure reproducibility we've uploaded the source code used to train models and make all the plots in the paper in supplementary materials. See the `readme.md` file at the root of the uploaded folder for reproducibility instructions. The `./notebooks` folder contains the jupyter notebooks for making plots. The datasets we've used are obtained from the code of respective papers (Gorishniy et al., 2025; Rubachev et al., 2025).

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

## A    UNCERTAINTY PLOTS FOR OTHER DATASETS

In this section we provide the uncertainty plots for all regression datasets from the Rubachev et al. (2025) and Gorishniy et al. (2021) benchmarks. In the main text we report the most informative plots that clearly illustrate our observations. The uncertainty plots for all models on all datasets are in Figure 13 and Figure 14.

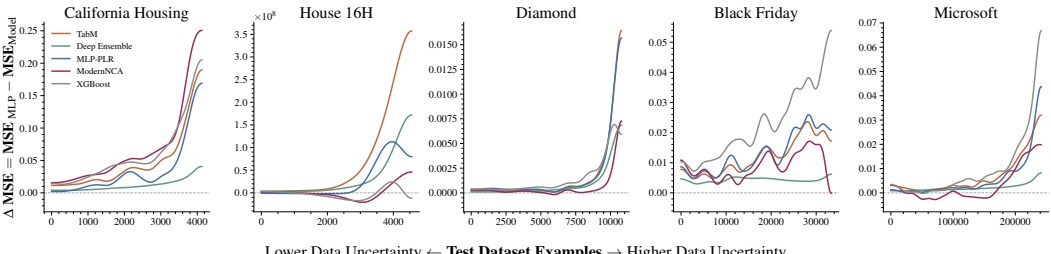

Figure 13: Uncertainty plots depicting differences between MLP and XGBoost, Deep Ensemble, MLP-PLR, ModernNCA and TabM.

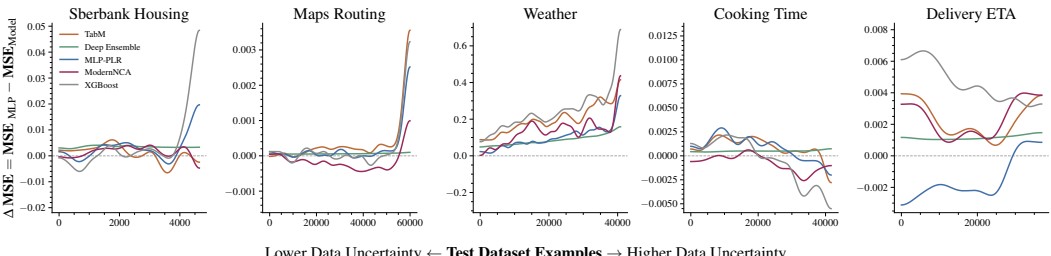

Figure 14: Uncertainty plots depicting differences between MLP and XGBoost, Deep Ensemble, MLP-PLR, ModernNCA and TabM. The Cooking Time dataset is an exception to our analysis, which we believe stems from the failure of data uncertainty estimation on this dataset.

## B    TECHNICAL DETAILS REGARDING UNCERTAINTY PLOTS

To estimate data uncertainty, we use CatBoost with the RMSEWithUncertainty loss function. Then, we sort the test datapoints in ascending order based on their estimated data uncertainty values, and calculate the difference in MSE between two models for each datapoint. Finally, we apply gaussian smoothing, using function gaussial_filter1d from scipy library.

## C    TECHNICAL DETAILS REGARDING HYPERPARAMETER TUNING FOR MODELS IN EMPIRICAL EXPERIMENTS

In this section we describe the hyperparameter spaces that were used to tune model hyperparameters in the experiments.

For the TabM model, we use the default TabM model from the reference implementation and follow Gorishniy et al. (2025) tuning spaces. For ModernNCA, MLP and MLP-PLR we also follow the tuning spaces from Gorishniy et al. (2025), except using powers of two for the neural network width tuning. For the exact tuning spaces and best parameters see the source code provided with the submission.

## D    TECHNICAL DETAILS REGARDING A SYNTHETIC SAW-LIKE DATASET

The data were generated as follows: objects $x_i$ are randomly and uniformly sampled from a two-dimensional rectangle with vertices $(0,0), (1,0), (1,10), (0,10)$. We then construct a saw-like separation line, to the left of which the clean targets $f_i$ are set to 1, and to the right of which the clean targets $f_i$ are 0, as shown in Figure 4. After that step, Gaussian noise with standard deviation $e^{g(x)} = \frac{x_2^6}{62500}$ is added to the targets, where $e^{g(x)}$ depends only on the vertical position of $x$ and grows as its value changes from 0 to 10, as also visualized on Figure 4. The final targets $y_i$ are produced as $y_i = f_i + e^{g(x_i)} \cdot \mathcal{N}(0,1)$. We use 80,000 training data points and 10,000 validation and test data points each. The exact algorithm used for the generation of the dataset is provided in the code. Generally speaking, we uniformly sample points from $(0,0)$ to $(1,10)$, and then set the target value as 1 inside the triangles with vertices $(2i, 0), (2i+1, 1), (2i+2, 0)$, for values of i: $0, 1, 2, 3, 4$. We also add the random normal noise to the targets, with the standard deviation of the noise set to $\frac{x_1^6}{4}$, for $x_1$ from 0 to 10.

## E    TECHNICAL DETAILS REGARDING HYPERPARAMETER TUNING FOR MODELS ON SYNTHETIC DATASETS

Here we describe the technical details regarding hyperparameter tuning for synthetic datasets.

**Saw-like synthetic data**. For this experiment we keep all model-architecture hyperparameters fixed at reasonable default values (3 layers, hidden dimension is 256, dropout is 0.2, 16 branches for TabM, dimension of embedding is 64 for MLP-LRLR) and only tune the learning rates, except the ModernNCA model for which we also tune the architecture hyperparameters. Details regarding tuning, early-stopping and other training protocols follow the one described in Appendix C.

**Synthetic data from the section 5 on numerical embeddings**. For this dataset we tune the MLP-PLR model using the same protocol that is described in Appendix C.

## F    TECHNICAL DETAILS REGARDING SYNTHETIC DATASET WITH $f(\cdot)$ AND $g(\cdot)$ PARAMETERIZED WITH MLPS

We generate this dataset as follows. First, we sample 40000 20-dimentional standard normal vectors. Then, we employ two MLPs to parametrize functions $f(\cdot)$ and $g(\cdot)$. The MLP simulating function $f(\cdot)$ consists of three linear layers separated by ReLU activations. The MLP simulating function $g(\cdot)$ consists of two linear layers separated by a ReLU activation, with hidden dimension being equal to 10. We generate $y_i$ as $f(x_i) + g(x_i) \cdot \mathcal{N}(0,1)$.

## G    DETAILS REGARDING THE MORE EFFECTIVE EMBEDDINGS LEARNING AND HYPERPARAMETER TUNING

The architecture of our module consists of the "-LRLR" embedder described in Gorishniy et al. (2022), followed by a single linear layer. We train the embedder module using the standard triplet loss (Schroff et al., 2015). More specifically, we first randomly sample a batch of training objects that will serve as anchors in the triplet loss. Then, for each anchor object, we randomly sample a pair of other training objects, of which the one with the closer target becomes a positive example, and the other one becomes the negative example. We then compute the embeddings for all three objects with our embedder as well as two dot-product similarities: the first one is between the embeddings of the anchor and the positive example, and the second one is between the embeddings of anchor and the negative example. We then pass these similarities as logits to the cross-entropy loss, for which the target is "0". By doing this, we effectively force the model to learn closer representations for objects with more similar targets.

After pretraining the embedder, we discard its last linear layer and use its remaining part to initialize the embedding part of the "MLP-LRLR" architecture. The tuning space for hyperparameters is presented in Table 3.

Table 3: The hyperparameter tuning space for the MLP and MLP-LRLR.

| Parameter | Distribution |
|---|---|
| # layers | $\text{UniformInt}[1, 4]$ |
| Width (hidden size) | $\text{PowersOfTwo}[2^7, 2^{11}]$ |
| Dropout rate | $\{0.0, \text{Uniform}[0.0, 0.75]\}$ |
| Weight decay[*] | $\{0, \text{LogUniform}[1e\text{-}6, 1e\text{-}3]\}$ |
| Learning rate[*] | $\text{LogUniform}[3e\text{-}5, 1e\text{-}3]$ |

\* – We decouple optimizer parameters for the triplet pretraining and finetuning phases.

**Embedding Parameters** (for LRLR and LRLR$_{\text{triplet}}$)

| | |
|---|---|
| d_embedding | $\{64, 128\}$ |
| # Tuning iterations | 100 |

The code for reproducing the results for the more effective embedding scheme with triplet-loss based pretraining is available in the supplementary materials. In **??** we provide all unaggregated results on the regression datasets from the Gorishniy et al. (2025) benchmark no larger than 50K samples. We also exclude two problematic datasets reported by Tschalzev et al. (2025). When computing ranks for the aggregations above we use code provided with the supplementary materials that uses standard deviations to account for the statistical significance of the ranking.

Table 4: Extended results for the main benchmark. Results are grouped by datasets.

| Ailerons ↓ | | | MiamiHousing2016 ↓ | | |
|---|---|---|---|---|---|
| Method | Single model | Ensemble | Method | Single model | Ensemble |
| MLP | $0.0002 \pm 0.0000$ | $0.0002 \pm 0.0000$ | MLP | $0.1604 \pm 0.0029$ | $0.1561 \pm 0.0026$ |
| MLP-LRLR | $0.0002 \pm 0.0000$ | $0.0002 \pm 0.0000$ | MLP-LRLR | $0.1507 \pm 0.0028$ | $0.1483 \pm 0.0032$ |
| MLP-LRLR$_{\text{triplet}}$ | $0.0002 \pm 0.0000$ | $0.0002 \pm 0.0000$ | MLP-LRLR$_{\text{triplet}}$ | $0.1471 \pm 0.0023$ | $0.1445 \pm 0.0018$ |

| OnlineNewsPopularity ↓ | | | analcatdata_supreme ↓ | | |
|---|---|---|---|---|---|
| Method | Single model | Ensemble | Method | Single model | Ensemble |
| MLP | $0.8635 \pm 0.0007$ | $0.8622 \pm 0.0003$ | MLP | $0.0781 \pm 0.0097$ | $0.0762 \pm 0.0090$ |
| MLP-LRLR | $0.8595 \pm 0.0010$ | $0.8568 \pm 0.0003$ | MLP-LRLR | $0.0775 \pm 0.0077$ | $0.0762 \pm 0.0080$ |
| MLP-LRLR$_{\text{triplet}}$ | $0.8581 \pm 0.0010$ | $0.8557 \pm 0.0005$ | MLP-LRLR$_{\text{triplet}}$ | $0.0774 \pm 0.0088$ | $0.0766 \pm 0.0093$ |

| california ↓ | | | cpu_act ↓ | | |
|---|---|---|---|---|---|
| Method | Single model | Ensemble | Method | Single model | Ensemble |
| MLP | $0.4915 \pm 0.0041$ | $0.4844 \pm 0.0023$ | MLP | $2.7117 \pm 0.1624$ | $2.5668 \pm 0.1102$ |
| MLP-LRLR | $0.4628 \pm 0.0024$ | $0.4536 \pm 0.0019$ | MLP-LRLR | $2.2692 \pm 0.0676$ | $2.1990 \pm 0.0698$ |
| MLP-LRLR$_{\text{triplet}}$ | $0.4582 \pm 0.0032$ | $0.4494 \pm 0.0011$ | MLP-LRLR$_{\text{triplet}}$ | $2.2782 \pm 0.1183$ | $2.2019 \pm 0.0979$ |

| diamond ↓ | | | elevators ↓ | | |
|---|---|---|---|---|---|
| Method | Single model | Ensemble | Method | Single model | Ensemble |
| MLP | $0.1395 \pm 0.0012$ | $0.1373 \pm 0.0003$ | MLP | $0.0020 \pm 0.0000$ | $0.0019 \pm 0.0000$ |
| MLP-LRLR | $0.1342 \pm 0.0013$ | $0.1324 \pm 0.0008$ | MLP-LRLR | $0.0018 \pm 0.0000$ | $0.0018 \pm 0.0000$ |
| MLP-LRLR$_{\text{triplet}}$ | $0.1328 \pm 0.0010$ | $0.1321 \pm 0.0004$ | MLP-LRLR$_{\text{triplet}}$ | $0.0018 \pm 0.0000$ | $0.0018 \pm 0.0000$ |

| fifa ↓ | | | house ↓ | | |
|---|---|---|---|---|---|
| Method | Single model | Ensemble | Method | Single model | Ensemble |
| MLP | $0.8025 \pm 0.0135$ | $0.8005 \pm 0.0149$ | MLP | $3.1026 \pm 0.0490$ | $3.0032 \pm 0.0062$ |
| MLP-LRLR | $0.7880 \pm 0.0114$ | $0.7849 \pm 0.0122$ | MLP-LRLR | $3.1181 \pm 0.0568$ | $3.0434 \pm 0.0354$ |
| MLP-LRLR$_{\text{triplet}}$ | $0.7863 \pm 0.0112$ | $0.7836 \pm 0.0124$ | MLP-LRLR$_{\text{triplet}}$ | $3.0705 \pm 0.0156$ | $3.0470 \pm 0.0078$ |

| house_sales ↓ | | |
| --- | --- | --- |
| Method | Single model | Ensemble |
| MLP | $0.1812 \pm 0.0009$ | $0.1781 \pm 0.0004$ |
| MLP-LRLR | $0.1677 \pm 0.0005$ | $0.1660 \pm 0.0002$ |
| MLP-LRLR$_{\text{triplet}}$ | $0.1696 \pm 0.0006$ | $0.1677 \pm 0.0001$ |

| isolet ↓ | | |
| --- | --- | --- |
| Method | Single model | Ensemble |
| MLP | $2.2740 \pm 0.3170$ | $2.0363 \pm 0.1421$ |
| MLP-LRLR | $2.3179 \pm 0.1404$ | $2.1286 \pm 0.0940$ |
| MLP-LRLR$_{\text{triplet}}$ | $2.3119 \pm 0.1506$ | $2.1873 \pm 0.1349$ |

| particulate-matter-ukair-2017 ↓ | | |
| --- | --- | --- |
| Method | Single model | Ensemble |
| MLP | $0.3779 \pm 0.0006$ | $0.3754 \pm 0.0002$ |
| MLP-LRLR | $0.3661 \pm 0.0009$ | $0.3634 \pm 0.0002$ |
| MLP-LRLR$_{\text{triplet}}$ | $0.3652 \pm 0.0006$ | $0.3629 \pm 0.0002$ |

| pol ↓ | | |
| --- | --- | --- |
| Method | Single model | Ensemble |
| MLP | $5.6147 \pm 0.6212$ | $5.1092 \pm 0.6127$ |
| MLP-LRLR | $2.6721 \pm 0.1627$ | $2.4205 \pm 0.1062$ |
| MLP-LRLR$_{\text{triplet}}$ | $2.5144 \pm 0.1259$ | $2.3130 \pm 0.0560$ |

| sberbank-housing ↓ | | |
| --- | --- | --- |
| Method | Single model | Ensemble |
| MLP | $0.2508 \pm 0.0046$ | $0.2447 \pm 0.0019$ |
| MLP-LRLR | $0.2427 \pm 0.0054$ | $0.2383 \pm 0.0014$ |
| MLP-LRLR$_{\text{triplet}}$ | $0.2405 \pm 0.0084$ | $0.2323 \pm 0.0013$ |

| superconduct ↓ | | |
| --- | --- | --- |
| Method | Single model | Ensemble |
| MLP | $10.7537 \pm 0.0778$ | $10.3687 \pm 0.0162$ |
| MLP-LRLR | $10.7919 \pm 0.1539$ | $10.3938 \pm 0.0130$ |
| MLP-LRLR$_{\text{triplet}}$ | $10.6695 \pm 0.0817$ | $10.3082 \pm 0.0451$ |

| wine_quality ↓ | | |
| --- | --- | --- |
| Method | Single model | Ensemble |
| MLP | $0.6682 \pm 0.0139$ | $0.6573 \pm 0.0153$ |
| MLP-LRLR | $0.6717 \pm 0.0155$ | $0.6510 \pm 0.0170$ |
| MLP-LRLR$_{\text{triplet}}$ | $0.6770 \pm 0.0212$ | $0.6608 \pm 0.0229$ |

| year ↓ | | |
| --- | --- | --- |
| Method | Single model | Ensemble |
| MLP | $8.9732 \pm 0.0237$ | $8.8923 \pm 0.0058$ |
| MLP-LRLR | $8.9511 \pm 0.0190$ | $8.9168 \pm 0.0059$ |
| MLP-LRLR$_{\text{triplet}}$ | $8.9398 \pm 0.0095$ | $8.9246 \pm 0.0042$ |

## H  DECOMPOSING AND COLLECTING MSE GRADIENT INTO CLEAN AND NOISY PART FOR A SYNTHETIC DATASET

First, we decompose the model's gradient for a particular datapoint into a "clean" component and a "noisy" component. We denote the model as a function $\phi(\cdot)$ and denote the model weights as $\theta$. As described in subsection 4.2, the target values $y_i$ are comprised of the noiseless part and the noisy term $y_i = f(x_i) + e^{g(x_i)} \cdot \mathcal{N}(0,1)$. Therefore,

$$
\frac{\partial(\phi(x_i) - y_i)^2}{\partial \theta} = 2(\phi(x_i) - y_i)\frac{\partial \phi(x_i)}{\partial \theta} = 2(\phi(x_i) - f(x_i) - e^{g(x_i)} \cdot \mathcal{N}(0,1))\frac{\partial \phi(x_i)}{\partial \theta} =
$$
$$
= 2(\phi(x_i) - f(x_i))\frac{\partial \phi(x_i)}{\partial \theta} - 2e^{g(x_i)} \cdot \mathcal{N}(0,1) \cdot \frac{\partial \phi(x_i)}{\partial \theta}
\tag{2}
$$

In this decomposition, the first term corresponds to the gradient of the noiseless part of the target dependency $f(\cdot)$. The second term corresponds to the noisy component of the gradient. Both terms can be explicitly computed for the synthetic data from subsection 4.2, since both $f(\cdot)$ and $g(\cdot)$ are known.

To collect the noisy and clean components of the gradients for TabM, its individual branches, and MLP, we first set both models to have the exact same hyperparameters, to avoid differences caused by difference in architecture. Then, we train both models until they both achieve performance close to the optimal, and collect gradients from test subset of the data. This ensures, that both models were properly trained, and the difference in gradient behavior is not caused simply by differences in initialization. Additionally, since we collect the gradients on test subset, we avoid overfitting to specific train examples. Therefore, we emulate a situation in which both models were already somewhat trained, but have not yet overfit to the data on which we conduct the experiment.

# I CONSISTENCY OF THE PROVIDED RESULTS WHEN USING DIFFERENT MODELS USED FOR DATA UNCERTAINTY ESTIMATION

In this section, we demonstrate that results we obtained in section 5, section 6 and section 7 stay the same when using different models for uncertainty estimation.

As can be seen on Figure 15, Figure 16, Figure 17, Figure 18, Figure 19, Figure 20, our analysis is not dependent on which specific model we use to estimate data uncertainty.

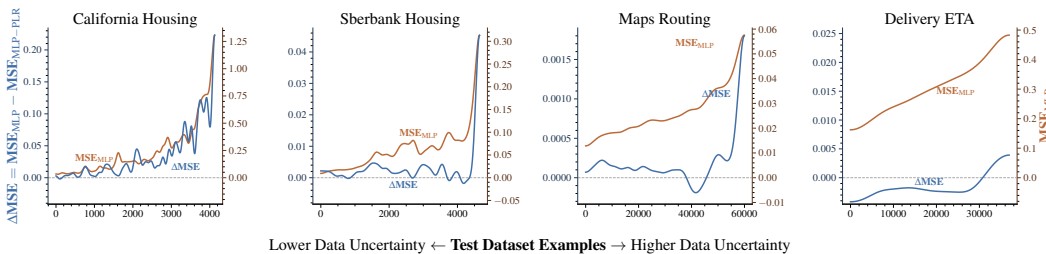

Figure 15: Version of Figure 6, but using MLP as a basic model for estimating data uncertainty. As can be seen, using different model for estimation leads to the same conclusion.

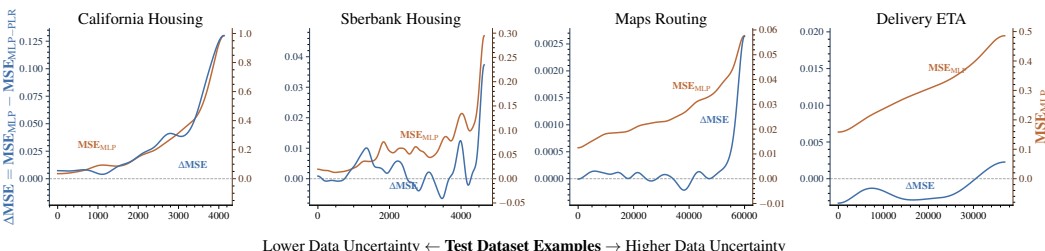

Figure 16: Version of Figure 6, but using MLP-PLR as a basic model for estimating data uncertainty. As can be seen, using different model for estimation leads to the same conclusion.

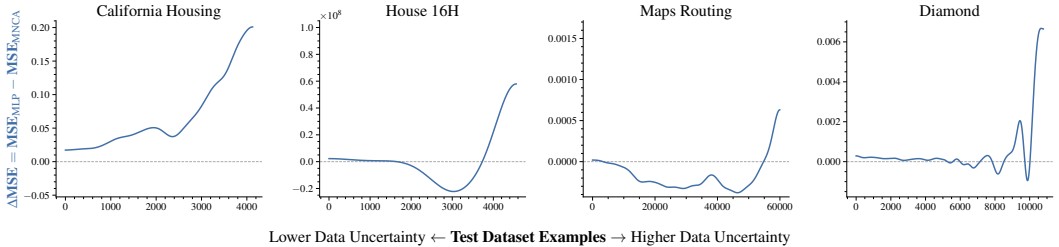

Figure 17: Version of Figure 8, but using MLP as a basic model for estimating data uncertainty. As can be seen, using different model for estimation leads to the same conclusion.

# J THE USE OF LARGE LANGUAGE MODELS

In the process of writing this paper, LLMs were only used for spellchecking, polishing writing and to aid in building cleaner plots. All the content, including reported experimental results and the data behind all plots, was obtained without using LLMs.

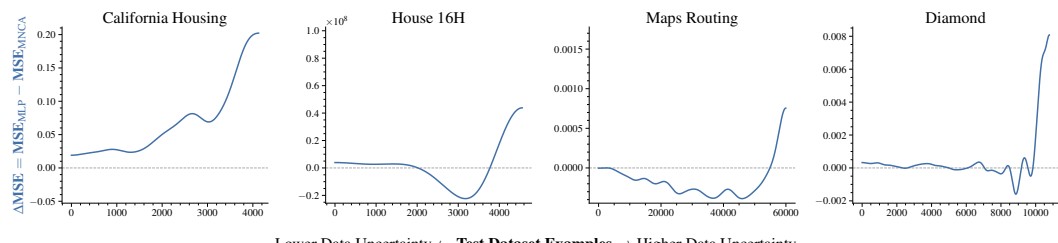

Figure 18: Version of Figure 8, but using MLP-PLR as a basic model for estimating data uncertainty. As can be seen, using different model for estimation leads to the same conclusion.

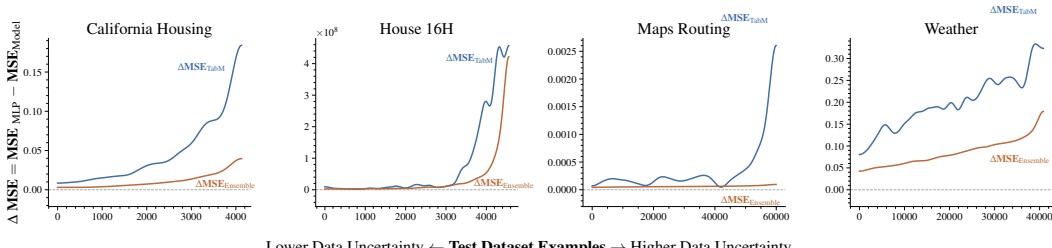

Figure 19: Version of Figure 10, but using MLP as a basic model for estimating data uncertainty. As can be seen, using different model for estimation leads to the same conclusion.

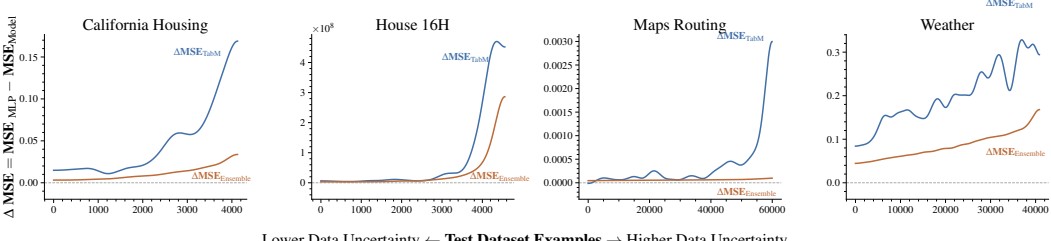

Figure 20: Version of Figure 10, but using MLP-PLR as a basic model for estimating data uncertainty. As can be seen, using different model for estimation leads to the same conclusion.

## K  UNCERTAINTY PLOTS FOR CLASSIFICATION DATASETS

In this section, we provide differences in accuracies between different methods, sorted by the entropy of predicted probabilities. These results are provided for the Adult, Higgs, and Uncertain CIFAR-10 datasets in Figure 21.

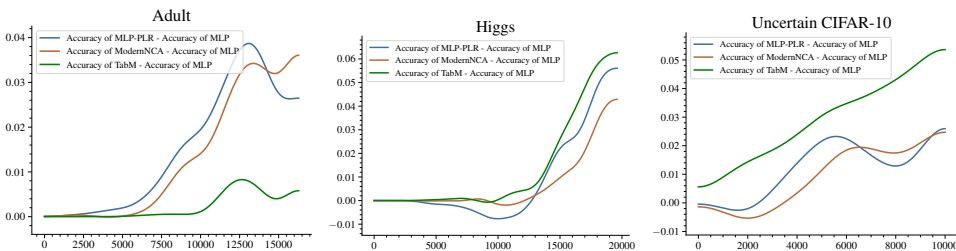

Figure 21: Differences in accuracy between several methods and MLP for the Adult, Higgs and Uncertain CIFAR-10 classification datasets, sorted by increasing data uncertainty left-to-right. Resulting plots are smoothed and averaged across all seeds.

## L  AN EXPERIMENT ON A SYNTHETIC DATASET WITH ASYMMETRIC NOISE

In this section, we describe an experiment, in which we generate a synthetic dataset following the same procedure as in subsection 4.1, but in addition to generating variance of the added noise distribution, second MLP also generates skew of said distribution. When generating the dataset, we first obtain clean targets through MLP $f(\cdot)$. To simulate label noise, we corrupt train and validation portions of the data, adding a noise with a skew-normal distribution which has zero mean and skew and variance generated by an MLP $g(\cdot)$, which is initialized independently from $f(\cdot)$. Figure 22 shows the distribution of skews used in the dataset.

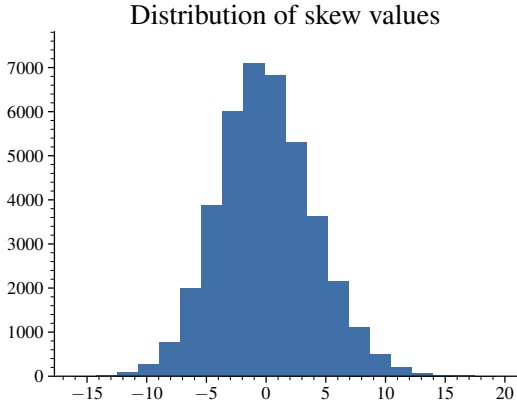

Figure 22: Distribution of values of skew in the added noise distribution.

As shown in Figure 23, even when the additive noise distribution is not Gaussian, our data uncertainty estimation procedure still performs highly.

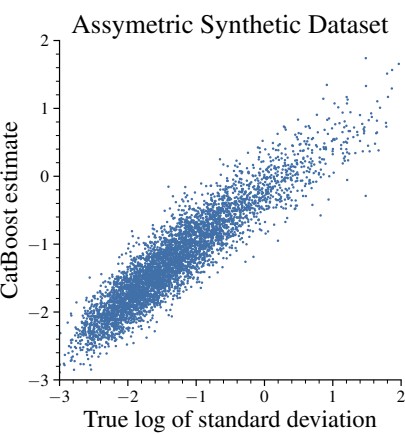

Figure 23: This figure demonstrates that our CatBoost-based procedure estimates the true log of data uncertainty with high precision.

Moreover, as shown in Figure 24, numerical embeddings, ModernNCA and TabM still show increased performance relative to MLP in the high data uncertainty niche.

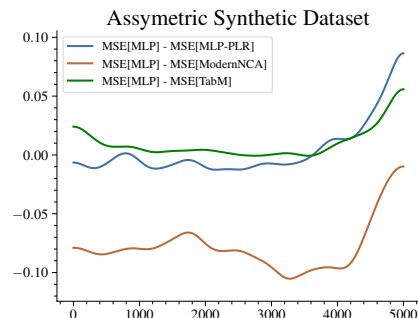

Figure 24: This figure shows an uncertainty plot like the one in Figure 3. As can be seen, our findings are confirmed even if the additive noise distribution is non-symmetric.

## M  AN EXPERIMENT ON THE VERSION OF THE CIFAR-10 DATASET WITH 50% LABEL NOISE.

In this section, we provide results of the studied methods on a version of CIFAR-10 dataset, where for 50% of train and validation samples the label was changed to a randomly and uniformly chosen different label. As shown in Table 5, MLP-LRLR begins to outperform MLP when both are trained on noisy data, while ModernNCA starts showing almost the same performance as MLP, although it is about a percentage worse when trained on the noiseless version of the dataset. While TabM and Deep Ensemble show the same accuracy when trained on the noiseless version of CIFAR, TabM outperforms Deep Ensemble when noise is injected in the training data.

Table 5: Accuracy of standard tabular DL methods on two versions of CIFAR-10. In each category (Single Models and Ensembles), the best method and methods with statistically insignificant differences from it are emphasized in bold.

|  | Accuracy ↑ | |
| --- | --- | --- |
| Method | CIFAR-10 | CIFAR-10 w/ 50% label noise |
| **Single Model Methods** | | |
| MLP | **58.31±0.27** | 46.28±0.44 |
| MLP-LRLR | 57.19±0.47 | **48.20±0.48** |
| ModernNCA | 57.48±0.22 | 46.20±0.60 |
| **Emsembling Methods** | | |
| Deep Ensemble | **60.92±0.22** | 48.02±0.27 |
| TabM | **61.01±0.25** | **48.63±0.27** |

## N  DIFFERENCE IN METRICS SORTED BY EPISTEMIC UNCERTAINTY ON MAPS ROUTING

Although aleatoric (data) and epistemic (knowledge) uncertainties often correlate, increases in performance of the studied methods correlate much cleanier with data uncertainty. To demonstrate this, we selected a dataset where that correlation is not particularly high, and show differences in MSE between MLP and other methods sorted by aleatoric and epistemic uncertainties (Figure 25). Both plots have the same scale and smoothing coefficient for clarity. As can be seen, the correlation of increased performance is much higher with data uncertainty than with knowledge uncertainty.

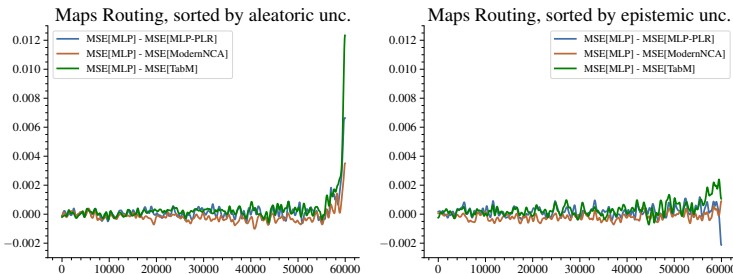

Figure 25: This figure shows an uncertainty plot like the one in Figure 3, but for the Maps-Routing dataset. We compare sorting based on aleatoric uncertainty and epistemic uncertainty.

## O  UNCERTAINTY PLOT EVOLUTION WITH ADDITIONAL NOISE

In this section, we study the behavior of MLP, TabM, and ModernNCA when a uniform standard normal noise is added to the dataset. As shown in Figure 26, all models degrade more on the parts of the dataset where data uncertainty is already high. Thus, even though the added noise is uniform, and the relative increase in noisiness is higher in the low data uncertainty niche, the advantage of methods like ModernNCA and TabM grows even more in the high data uncertainty niche. We conjecture that all models can overcome some level of noise in the training labels, but after some value, larger noise leads to a super-linear increase in MSE of the trained model.

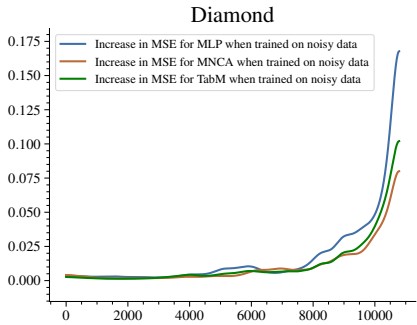

Figure 26: This figure shows increases in MSE when a model is trained on the noisy version of the dataset for different models. Results are sorted by data uncertainty, smoothed and averaged across all seeds.

## P  UNCERTAINTY PLOTS FOR TABPFN AND REALMLP MODELS

In this section, we study the behaviour of RealMLP (Holzmüller et al., 2024) and in-context learning based tabular foundation models (TabPFNv2 Hollmann et al. (2025) and TabPFNv2.5 Grinsztajn et al. (2025)) through the lens of data uncertainty.

We use the tuned RealMLP model with the default search space (50 random HPO trials) from pytabkit[4]. For both TabPFN models we use the default configuration with `n_estimators` set to one, to eliminate effects from ensembling different model configurations. Uncertainty plots sor TabPFN models, RealMLP and ModernNCA (for reference comparison) are in Figure 27. We can see that tabular foundation models and RealMLP also improve performance on samples with high data uncertainty. Furthermore, the inverse is also true, on the House dataset where both TabPFNv2 and RealMLP do not outperform the MLP baseline, this dynamic is most clear on high data uncertainty samples.

---

[4] https://github.com/dholzmueller/pytabkit

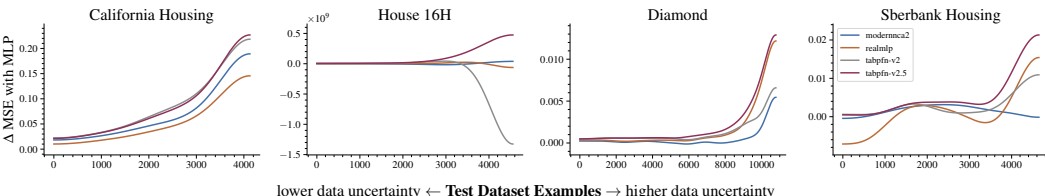

Figure 27: An uncertainty plot, comparing various additional models in TabPFNv2, TabPFNv2.5 and RealMLP to the MLP baseline. We can see that these models too, improve performance in the high data uncertainty niche.

## Q  THE ROLE OF WEIGHT DECAY AND DROPOUT HYPERPARAMETERS IN IMPROVING PERFORMANCE ON HIGH DATA UNCERTAINTY EXAMPLES

In this section we study whether the hyperparameters like weight decay or dropout play a role in managing data uncertainty. We re-ran the MLP model with these regularization methods explicitly disabled. The original MLP configuration already includes both weight decay and dropout as optional tuned hyperparameters. In the new experiment, we compare three variants: standard MLP, MLP without weight decay and MLP without dropout. The results in Figure 28 reveal no consistent pattern across uncertainty levels. On some datasets, disabling dropout slightly improves performance regardless of uncertainty, on others it degrades results, on a subset of datasets there may be a slight dependency. Weight decay shows even weaker effects, with score differences relative to the MLP averaging around -0.19% (±0.31%). The absence of a clear trend specific to high-uncertainty regions (unlike the strong patterns observed with TabM or numerical embeddings) suggests these general-purpose hyperparameters like weight decay or dropout do not specifically target data uncertainty.

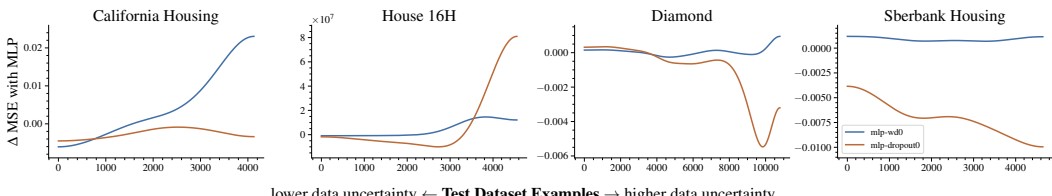

Figure 28: An uncertainty plot, comparing various additional models in TabPFNv2, TabPFNv2.5 and RealMLP to the MLP baseline. We can see that these models too, improve performance in the high data uncertainty niche.

## R  UNCERTAINTY PLOTS ON DATASETS FROM THE TABARENA BENCHMARK

We also run experiments on datasets from the TabArena benchmark. For this we take TabArena regression datasets wiht more than 10,000 samples, and tune MLP, MLP-PLR, TabM and ModernNCA models on these datasets (we use hold-out valiadation for the tuning procedure). To construct the uncertainty plots we use median as the window smoother to filter outliers, as TabAarena datasets are smaller relative to other datasets present in our analysis (which makes it harder to see the sample-wise dynamic and increases the impact of outliers). As we can see in Figure 29 the trend of increased improvement on higher data uncertatinty holds on TabArena datasets, all three method (TabM, ModernNCA and PLR embeddings) tend to improve perfomance more on examples with high data unceratainty.

## S  UNCERTAINTY PLOTS FOR MODELS OF DIFFERENT CAPACITY

In this section we explore how does model size/capacity affect our findings. We tune and evaluate models at three model sizes for MLP and TabM models on four datasets (California Housing, House 16H, Diamond and Sberbank Housing). We vary the model size manually and have three increasing

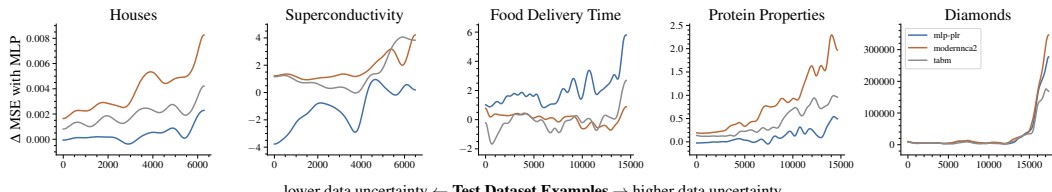

Figure 29: An uncertainty plot, comparing various additional models in TabPFNv2, TabPFNv2.5 and RealMLP to the MLP baseline. We can see that these models too, improve performance in the high data uncertainty niche.

variations (1 hidden layer of size 128, 2 hidden layers of size 256 and 4 hidden layers of size 512). We use the 1 layer 128 wide MLP as a reference model to make uncertainty plots. In Figure 30 we show uncertainty plots for MLP models of different capacity. We can see that when model capacity increases models tend to improve on high data uncertainty examples – this is an interesting avenue for future research into how model capacity can interract with data uncertainty during model training. We hypothesize that increasing model size but keeping all other regularization (such as early stopping, weight decay, dropout) could help make models more robust to noisy examples in training.

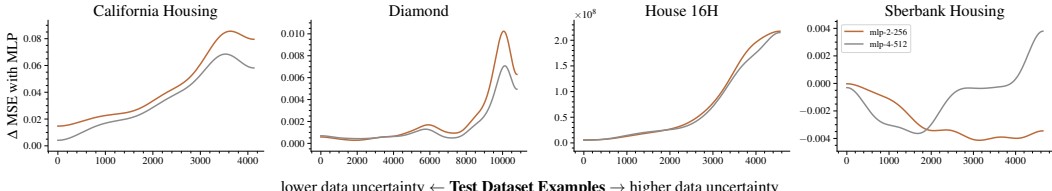

Figure 30: An uncertainty plot, comparing MLP's of increasing size to the smallest 1 hidden layer 128 unit MLP. Interestingly, increasing model capacity helps more on high data unceratinty samples.

We also run a similar experiment with the TabM model, we tune and evaluate TabM models of different capacities and compare those to the smallest MLP model. The results in Figure 31 provide two findings. First, the smallest TabM achieves small to no improvement compared to the MLP (which aligns with observations of TabM models being slightly bigger on average reported in the TabM paper (Gorishniy et al., 2025)). Second, with suffcent capacity TabM improvements are seen on high data uncertainty examples as we've observed before.

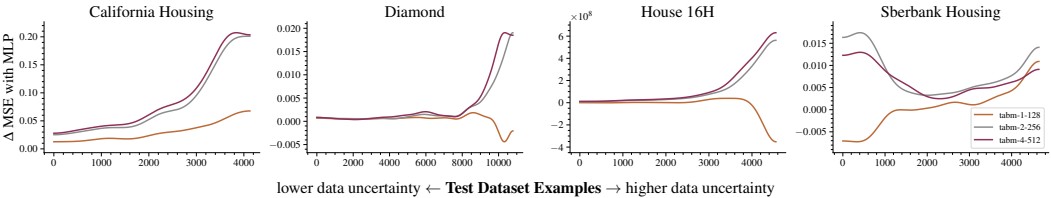

Figure 31: An uncertainty plot, comparing TabM models of increasing size to the smallest 1 hidden layer 128 unit MLP.

## T   RESULTS VARIABILITY ON DIFFERENT SYNTHETIC UNCERTAIN CIFAR-10 VARIATIONS

In this section we demonstrate that our results from Table 1 stay the same regardless of the seed we've used to create the Uncertain CIFAR-10 dataset. We repeat the tuning and evaluation of the four models presented in Table 1 with different dataset preparation random seeds and report results in Table 6. We can see that results align very closely with those presented in Table 1. We can see that on uncertain CIFAR MLP-LRLR and ModernNCA provide certain improvements over MLP regardless

of the seed used to make a dataset. For the ensemble methods TabM improves upon the regular deep ensemble on all uncertain CIFAR variations, and does not improve on the regular CIFAR-10.

Table 6: Accuracy of standard tabular DL methods on CIFAR-10 and multiple Uncertain CIFAR-10 seed variations. Results for CIFAR-10 and Seed 0 are from the original experiments; Seeds 1–4 are additional runs. Bold indicates methods whose mean−std exceeds competitors' means.

| | | Uncertain CIFAR-10 | | | | |
|---|---|---|---|---|---|---|
| Method | CIFAR-10 | Seed 0 | Seed 1 | Seed 2 | Seed 3 | Seed 4 |
| **Single Model Methods** | | | | | | |
| MLP | **58.31±0.27** | 47.54±0.17 | 47.85±0.29 | 46.94±0.30 | 47.62±0.25 | 47.42±0.33 |
| MLP-LRLR | 57.19±0.47 | **48.73±0.23** | 47.82±0.25 | **47.73±0.25** | **48.10±0.14** | 48.00±0.25 |
| ModernNCA | 57.48±0.22 | 48.45±0.30 | **48.51±0.34** | 47.49±0.20 | **48.02±0.29** | **48.72±0.31** |
| **Ensemble Methods** | | | | | | |
| Deep Ensemble | 60.92±0.22 | 49.41±0.20 | 48.48±0.23 | 47.59±0.05 | 48.24±0.17 | 48.37±0.34 |
| TabM | 61.01±0.25 | **50.45±0.19** | **50.14±0.00** | **48.84±0.00** | **49.97±0.17** | **50.04±0.24** |