# OpenReview forum: "Unveiling the Role of Data Uncertainty in Tabular Deep Learning"
_ICLR.cc/2026/Conference — Submitted to ICLR 2026_

### Official Review · Reviewer_nGFb · 2025-10-17

**Soundness:** 3
**Presentation:** 3
**Contribution:** 3
**Rating:** 6
**Confidence:** 4

**Summary:**

The paper proposes a unifying, sample-wise lens—data (aleatoric) uncertainty—to explain why several recent choices in tabular DL tend to help. Using per-example uncertainty (estimated via a probabilistic CatBoost fit), the authors draw “uncertainty plots” that sort test samples by estimated uncertainty and compare MSE deltas between baselines (MLP) and—(i) numerical embeddings, (ii) retrieval-augmented models (ModernNCA), and (iii) parameter-efficient ensembling (TabM). The key empirical claim is that these techniques help disproportionately more on the high-uncertainty slice. This framing also motivates a small practical contribution: a triplet-trained numerical embedding that modestly improves over LRLR embeddings across multiple datasets.

**Strengths:**

Clear, unifying viewpoint that organizes disparate tabular DL results under one explanatory factor (data uncertainty). The uncertainty-slice analysis feels insightful and actionable.

Careful sanity checks: synthetic datasets with known uncertainty; robustness of findings to different uncertainty estimators (Appendix I).

Mechanistic probes beyond benchmarks (e.g., neighbor-target consistency, gradient clean/noisy decomposition for TabM).

Breadth: results cover several public tabular benchmarks; includes a pruned “Uncertain CIFAR-10” to stress uncertainty effects across settings.

Reproducibility: code and notebooks promised; tuning spaces documented.

**Weaknesses:**

The core plots hinge on CatBoost RMSEWithUncertainty estimates and Gaussian smoothing; while Appendix I shows estimator-robustness, the absolute calibration of uncertainty remains unvalidated on real data beyond rank correlations seen on synthetic tasks. Consider evaluating calibration (e.g., probability-integral transforms / reliability curves for σ̂(x)) and reporting agreement across multiple families (GBDT, MLP-heteroscedastic heads, NGBoost).

Uncertainty plots are compelling but largely visual. Provide confidence bands or per-quantile paired tests (e.g., Wilcoxon on per-example ΔMSE within quantile bins) to quantify “disproportionate” improvements, and report effect sizes across datasets.

Focus is mostly regression; classification appears only via Uncertain CIFAR-10 with 50 features retained (Table 1). Stronger support on standard tabular classification (incl. class-imbalance) would broaden impact.

The paper explicitly does not analyze epistemic uncertainty or generic regularizers (dropout, weight decay, SAM), though these interact with label noise and could compete with retrieval/ensembling.

Gains over LRLR are modest (Δ rank and Δ% vs MLP are small), and details like triplet mining strategy, margin, and overhead are not fully explored. Please expand comparisons vs other metric-learning or contrastive objectives (e.g., supervised contrastive, NCA-style losses) and quantify cost/benefit.

**Questions:**

Compare CatBoost RMSEWithUncertainty vs heteroscedastic MLP, NGBoost, CatBoost ensembles; evaluate correlation and calibration on real datasets (e.g., split-and-replicate labels when possible).
Vary noise and show how ModernNCA/TabM curves move; verify TabM’s gradient noise reduction in practice with loss-landscape / gradient-norm probes.
Add standard classification datasets (adult, higgs, covertype, etc.) and relate ECE/NLL improvements to uncertainty. (You already include Uncertain CIFAR-10.)
Test if triplet pretraining helps ModernNCA backbones specifically.

---

> ### Author Response · Authors · 2025-11-25
>
> Thank you for the review.
>
> > The absolute calibration of uncertainty remains unvalidated on real data beyond rank correlations seen on synthetic tasks.
>
> The task of validating data uncertainty on real data is challenging. For example, let us consider a synthetic dataset from subsection 4.1 of our paper; even on the clean test data, model error remains well above zero. Therefore, on real datasets, MSE on the test data will reflect the sum of irreducible loss due to data uncertainty and reducible gap in learning caused by training on noisy data [1].
>
> > Compare CatBoost RMSEWithUncertainty vs heteroscedastic MLP, NGBoost, CatBoost ensembles; evaluate correlation and calibration on real datasets (e.g., split-and-replicate labels when possible).
>
> Figure 2.a) in our paper shows high correlation between CatBoost RMSEWithUncertainty and heteroschedastic MLP on three real datasets. NGBoost uses the same data uncertainty estimation procedure as is used in our CatBoost realization, while CatBoost ensembles represent epistemic uncertainty instead of aleatoric uncertainty.
>
> > Vary noise and show how ModernNCA/TabM curves move; verify TabM’s gradient noise reduction in practice with loss-landscape / gradient-norm probes.
>
> In Appendix O, we study sample-wise changes in performance for ModernNCA, TabM and MLP for the diamond dataset, to which we add uniform normal noise. We visualize them by plotting increases in MSE for MLP, ModernNCA and TabM in Figure 26 of the revised manuscript when trained on this noisy version of the dataset in comparison with the baseline trained on the cleaner version. Interestingly, all methods (baseline MLP, TabM and ModernNCA) deteriorate the most in the part of the dataset for which the uncertainty already was the highest, while for the region of the data with the lowest data uncertainty, all models combat added noise in a highly effective manner. This could signal that a model's performance degrades faster than the level of noise, with models being able to ignore relatively low noise amplitudes. For practical illustration of TabM's mechanism of noise reduction, we show reduction of gradient norms in Figure 12.
>
> > Focus is mostly regression; classification appears only via Uncertain CIFAR-10 with 50 features retained (Table 1). Stronger support on standard tabular classification (incl. class-imbalance) would broaden impact.
>
> > Add standard classification datasets (adult, higgs, covertype, etc.)
>
> In Appendix K, we demonstrate uncertainty plots for adult, higgs, and Uncertain CIFAR-10 datasets. Entropy of predicted probabilities serves as a measure of data uncertainty in these experiments. HPO of studied methods on covertype dataset failed to converge in time for our response due to its large size. In our experiments, we found that the observed trends still hold for the task of classification.
>
> > The paper explicitly does not analyze epistemic uncertainty
>
> We found that on some datasets, epistemic and aleatoric uncertainties show some correlation. In that case, increases in performance correlate with epistemic uncertainty as well. However, on the datasets where knowledge (epistemic) and data (aleatoric) uncertainties do not correlate, such as maps-routing dataset, we only observe a relation between increased performance and data uncertainty. In Appendix N, we show differences in performance sorted by epistemic uncertainty instead of the aleatoric uncertainty on the maps-routing dataset.
>
> > or generic regularizers (dropout, weight decay, SAM)
>
> In Appendix Q, we provide uncertainty plots for weight decay and dropout, and find that their utility in combatting high data uncertainty is inconsistent across datasets. The robustness of SAM to noisy labels was already studied in the general case [2], and therefore we do not cover it in this paper.
>
> > though these interact with label noise and could compete with retrieval/ensembling.
>
> All of the DL methods studied in our paper (including MLP) already include dropout and weight decay.
>
> > Gains over LRLR are modest (Δ rank and Δ% vs MLP are small),
>
> Relative improvement over MLP is 36% higher for the proposed embedding scheme than for baseline LRLR embeddings. We aknowledge that changes to some of our design choices possibly could increase this gap, but we treat this proposed embedding scheme as a proof of concept of the usefulness of our analysis, and therefore do not fully explore optimal design choices for this method.
>
> [1] Learning from Noisy Labels with Deep Neural
> Networks: A Survey, Song et al., https://arxiv.org/pdf/2007.08199
>
> [2] Why is SAM Robust to Label Noise? Baek et al., https://arxiv.org/abs/2405.03676

---

### Official Review · Reviewer_ixYt · 2025-10-28

**Soundness:** 3
**Presentation:** 3
**Contribution:** 3
**Rating:** 6
**Confidence:** 3

**Summary:**

This paper analyzes the correlation between _data uncertainty_ and the success of recent tabular deep learning models. The authors first start by demonstrating that XGBoost shows much stronger performance in high-uncertainty regions, and that recent tabular DL models start showing similar trends. Then the authors examine the performance of DL tabular models, on two synthetic datasets as well a suite of real-world datasets. The authors also propose an improvement to an existing tabular embedding model by incorporating uncertainty-reducing contrastive loss by applying the finding that embeddings "higher quality neighborhoods" lead to better uncertainty mitigation. Overall, this paper provides a set of results that help explain the success of recent tabular DL models and how the gap between GBDTs may be tackled.

**Strengths:**

- This paper presents an interesting analysis that reveals the connection between data uncertainty and the success of existing SotA tabular models. I think this is a very important contribution to the field, given that this same discussion has been brought up multiple times (WhyTrees, Tabzilla) but not fully explored. The findings from this paper seems to pinpoint a solid factor in the success of GBDT models, and also provide a path forward for tabular DL models to close the gap.
- The experiments are well designed and the results are clearly presented. The sawtooth dataset and the decision boundary are a nice touch for visualizing the effect of uncertainty.
- In addition to the analysis, the authors also propose a simple improvement to one of the analyzed methods that show consistent improvement.

**Weaknesses:**

- This paper only focuses on regression datasets. While this is understandable given the uncertainty estimation method, it would be nice to see if 1) can we have a similar notion of uncertainty for classification datasets, and 2) do the same trends hold.
- The entire analysis is based on the uncertainty-detection described in section 3.1. The authors do validate the robustness (and consistency) of this method, but the findings are contingent on this specific uncertainty estimation method.
- I found the figures with uncertainty in the x axis a bit difficult to follow at first. A potential suggestion here is to normalize the uncertainty values per dataset (divide by max) so that everything appears on the same scale.

**Questions:**

- Are there examples we can see of high-uncertainty sample vs. a low-uncertainty sample from the same dataset? That would be very nice.
- What exactly is the difference between -LRLR and -PLR? Are they both using the scheme from Gorshniy et al. 2022?
- For the contrastive loss, how are the positive and negative samples selected? Given that the datasets are regression datasets, is it some thresholding? And is it weighted?
- How does this new embedding method compare against other models? like ModernNCA or TabM? And/or can it be integrated into those models as well?
- The paper starts off by showing that GBDTs perform strongly in high-uncertainty regions. But then the rest of the paper focuses on tabular DL models. Are there more insights into what makes GBDTs so effective in these regions? Is it just the model capacity, or is there something more fundamental about the tree structure that helps?

---

> ### Author Response · Authors · 2025-11-25
>
> Thank you for your review.
>
> > This paper only focuses on regression datasets. While this is understandable given the uncertainty estimation method, it would be nice to see if 1) can we have a similar notion of uncertainty for classification datasets, and 2) do the same trends hold.
>
> While data uncertainty is not a phenomenon exclusive to regression tasks, when estimating data uncertainty for clasification tasks, finding a correct calibration technique provides an additional degree of freedom which could complicate analysis. Nevertheless, we believe increased data uncertainty in classification tasks also leads to increased performance in studied methods.
>
> The first way we demonstrate it, is by the synthetic experiment described in subsection 4.3. We produce a version of CIFAR-10 dataset that has increased data uncertainty, and demonstrate that the studied methods show less degradation in accuracy compared to MLP. Thus, we observe the studied effect even on the task of classification.
>
> Motivated by yours and other reviewers concerns, we demonstrate applicability of our findings to the task of classification in a second way and expand the study of classification problems in our manuscript:
>
> 1. In Appendix K, we provide uncertainty plots for popular classification datasets such as Adult and Higgs, as well as for our Uncertain CIFAR-10 dataset. We use entropy of predicted probabilities as a measure of data uncertainty, following [1].
> 2. In Appendix M, we provide the experiment on the version of CIFAR-10 with original features, but 50% symmetric label noise (in 50% of samples targets are changed to a random different label). We show that compared to MLP, results greatly improve on the noisy version of the dataset. The table with accuracy difference between MLP and studied methods is included below as ``Table 1``.
>
>
> **Table 1.** Accuracy of different methods on original and noisy versions of the CIFAR-10 datasets. MLP-LRLR begins to outperform MLP when both are trained on noisy data, while ModernNCA starts showing almost the same performance as MLP, although it is about a percentage worse when trained on the noiseless version of the dataset. While TabM and Deep Ensemble show the same accuracy when trained on the noiseless version of CIFAR, TabM outperforms Deep Ensemble when noise is injected in the training data.
>
> | Method | CIFAR-10 | CIFAR-10 with 50% label noise|
> | --- | --- | --- |
> | MLP | 58.31+-0.27 | 46.28+-0.44|
> | MLP-LRLR | 57.19+-0.47 | 48.20+-0.48|
> | ModernNCA | 57.48+-0.22 | 46.20+-0.60|
> | Deep Ensemble | 60.92+-0.22 |48.02+-0.27|
> | TabM | 61.01+-0.25 | 48.63+-0.27|
>
> We believe additional experiments described above allow our work to demonstrate applicability of our findings to the task of classification.
>
> > Are there examples we can see of high-uncertainty sample vs. a low-uncertainty sample from the same dataset? That would be very nice.
>
> In California Housing dataset, in general, high data uncertainty samples tend to be from the cities with high density, while low data uncertainty samples tend to be from the rural areas, which could be explained by rural housing market being less vulnerable to rapidly changing trends and less neighborhood-by-neighborhood variance.
>
> Another interesting example comes from the Sberbank Housing dataset, where some of the appartments come directly from a building company, while some are being resold by their previous inhabitants. The second category of units generally shows much greater data uncertainty, since previous owners may have modified an appartment in a way that would not necessarily be reflected by the features in the data but may yet influence the final selling price.
>
> > What exactly is the difference between -LRLR and -PLR? Are they both using the scheme from Gorshniy et al. 2022?
>
> They are both different embedding schemes proposed by the paper. PLR consists of a periodic layer (each feature x_i is replaced by vector of (sin(phi_0_i * x_i), cos(phi_0_i * x_i), sin(phi_1_i * x_i), cos(phi_1_i * x_i), ..., sin(phi_n_freq_i * x_i), cos(phi_n_freq_i * x_i)), where each phi_j_i is a learned parameter), followed by a linear layer and ReLU activation. LRLR consists of linear layer with 1 input feature and d output features, followed by ReLU, linear layer with d input and output features, and finally another ReLU activation. Both embeddings scheme build a learnable function between a feature and a vector, and both embedding schemes do not share any parameters between different features. Which embedding scheme is better varies between datasets, and we have not found any uncertainty-related differences in their behaviour.

---

> > ### Author Response · Authors · 2025-11-25
> >
> > > For the contrastive loss, how are the positive and negative samples selected? Given that the datasets are regression datasets, is it some thresholding? And is it weighted?
> >
> > For each object we randomly sample a pair of other examples, and then the one with closer target becomes a positive sample and the one with further target becomes a negative sample. While this sampling procedure could potentially be improved, we treat this new embedding pretraining scheme as a proof-of-concept for usefulness of uncertainty-guided analysis and not as the most perfect realization of our idea. Improving our embedding pretraining procedure is an interesting future work avenue for us.
> >
> > > The paper starts off by showing that GBDTs perform strongly in high-uncertainty regions. But then the rest of the paper focuses on tabular DL models. Are there more insights into what makes GBDTs so effective in these regions?
> >
> > When obtaining a prediction through a GBDT model, one effectively averages target values of samples that fall into the same "leaves" in the constituent trees of the model. This links GBDT to retrieval-based methods like ModernNCA, where prediction is obtained through averaging target values of train samples closest in a shared latent space. We conjecture that this similarity could be a possible reason behind GBDTs' increased robustness to high data uncertainty.
> >
> >
> > [1] Uncertainty in Gradient Boosting via Ensembles, Malinin & Ustimenko & Prokhorenkova, https://arxiv.org/pdf/2006.10562
> >
> > [2] TabArena: A Living Benchmark for Machine Learning on Tabular Data, Erickson et al., https://arxiv.org/abs/2506.16791
> >
> > [3] TabReD: Analyzing Pitfalls and Filling the Gaps in Tabular Deep Learning Benchmarks, Rubachev & Kartashev et al., https://proceedings.iclr.cc/paper_files/paper/2025/hash/571799482291411607c54984153190b0-Abstract-Conference.html

---

### Official Review · Reviewer_ewbA · 2025-10-29

**Soundness:** 2
**Presentation:** 3
**Contribution:** 1
**Rating:** 2
**Confidence:** 4

**Summary:**

This paper investigates why certain design choices in tabular deep learning—such as numerical embeddings, retrieval-augmented architectures, and model ensembling—often outperform standard MLPs. The authors propose that these methods are particularly effective in regions of high data uncertainty and develop a framework to quantify such uncertainty using heteroscedastic Gaussian modeling. Through experiments on both synthetic and real-world datasets, they show that the relative advantage of these architectures grows with estimated noise levels. The study provides a novel perspective linking model design and data uncertainty in tabular learning.

**Strengths:**

The paper provides a clear and well-motivated analysis of why certain design choices in tabular deep learning, such as embeddings, retrieval augmentation, and ensembling, might succeed. Framing these differences through the lens of data uncertainty offers a fresh and conceptually coherent perspective.

The paper is well written, with a clear motivation and logical flow.

**Weaknesses:**

1. **Narrow treatment of data uncertainty**

The paper assumes that all uncertainty arises from Gaussian noise and evaluates only this setting. Other realistic forms of noise—such as asymmetric, heavy-tailed, missing-feature, or label noise—are not considered. Consequently, it remains unclear whether the observed model advantages genuinely reflect robustness to data uncertainty in general or merely to this specific assumption. The conclusions may not hold once the Gaussian hypothesis is relaxed.

2. **Missing comparisons with strong modern baselines**

The experiments omit several state-of-the-art tabular models, notably TabPFN v1/v2 and ExcelFormer. TabPFN has shown outstanding performance across diverse benchmarks, while ExcelFormer claims improved robustness to noisy inputs in its experiments.

3. **Limited realism of real-world evaluations**

Although several real-world datasets are used (e.g., California Housing), they are relatively clean public benchmarks with minimal noise and well-structured features. The study does not consider genuinely messy or uncurated data, where uncertainty is more complex and multifactorial, particularly in domains such as healthcare or finance. Evaluating the approach on raw, noisy real-world datasets rather than sanitized benchmarks would provide a more compelling demonstration of how the proposed approach handles real noise in practice.

**Questions:**

See above.

---

> ### Author Response · Authors · 2025-11-25
>
> Thank you for your review.
>
> ## Weakness 1. Narrow treatment of data uncertainty
>
> > The paper assumes that all uncertainty arises from Gaussian noise and evaluates only this setting.
>
> When we refer to that assumption in the beginning of subsection 3.1, we only describe a mathematical model of the data, for which the loss used in our data uncertainty estimation is the true negative log-likelihood.
> Similar assumptions have often been used to introduce Machine Learning methods, for example, in a classical machine learning textbook ''Pattern Recognition and Machine Learning'' by Christopher Bishop [1], the same assumption is made when introducing the widely used MSE loss function:
>
> >"Thus the sum-of-squares error function has arisen as
> a consequence of maximizing likelihood under the assumption of a Gaussian noise distribution"
>
> Nevertheless, nowadays MSE function is used ubiquitously even on the data where the noise in the data is non-Gaussian. Similarly, our findings are valid in the settings where label noise does not follow a simple gaussian distribution.
>
> To demonstrate this, we have conducted a series of additional experiments:
>
> 1. In Appendix L, we conduct experiment on a synthetic dataset with **assymetric** noise. The dataset is akin to the one introduced in 4.1, but with skew-normal noise distribution used instead of just normal. The skew coefficient ranges from -15.97 to 19.22 and is depicted in the Appendix as well. Our results show not only that our estimation procedure still produces results highly correlated to the log of variance of the noise distribution, but that the effect of robustness to data uncertainty is still observed even when noise is not symmetric.
> 2. In Appendix M, we conduct experiments on the version of CIFAR-10 dataset which 50% **label noise**, and demonstrate that the studied methods show improved difference in performance to the MLP with added label noise.
> 3. In Appendix K, we show that **on the task of classification**, when data uncertainty is estimated as the entropy of the predicted probabilities and noise does not follow gaussian distribution, our findings still apply. Moreover, in the case of Uncertain CIFAR-10, this also demonstrates robustness to data uncertainty which arises from **missing features**.
>
> Above, we demonstrated that our data uncertainty estimation works even when the noise distribution is not Gaussian. In many of the datasets already studied in our paper, data uncertainty arises from sources other than Gaussian label noise. Our experiments with Uncertain CIFAR-10 show the impact of data uncertainty caused by missing features on relative performance of the studied methods. In datasets from the TabReD benchmark [2], many features have missing values, and in other datasets such as ''California Housing'' and ''House'' there are outliers in the data which cause the conditional probability distribution p(y | x) to be assymetric and heavy-tailed.
>
>
> ### Summary of our response to the Weakness 1.
>
> To **summarize**, our work is not limited to Gaussian noise. We believe that the experiments provided above, as well as our comments, demonstrate applicability of our findings to every potential source of data uncertainty you mention in your review.
>
> ## Weakness 2, Missing comparisons with strong modern baselines
>
>
> ### Tabular Foundational Models
>
> Motivated by your suggestion, we study robustness to data uncertainty of TabPFNv2 and TabPFNv2.5. In Appendix P, we include uncertainty plots for TabPFNv2 and TabPFNv2.5. These plots demonstrate that Foundational Tabular Models show robustness to high data uncertainty, which could be potentially explained by their relationship with retrieval methods([3]).
>
> ### RealMLP
>
> RealMLP [4] is a Tabular DL method that has shown great performance on a recent tabular benchmark TabArena [5]. To widen the scope of our paper, we decided to add this method to our study. In Appendix P, we provide uncertainty plots for RealMLP. These plots demonstrate that RealMLP shows robustness to high data uncertainty, which could be explained by both the advanced embedding scheme utilized in RealMLP as well as improvements to optimization which lead to more stable optimization in the precense of noise.
>
> ### ExcelFormer
>
> A recent paper introducing state-of-the art Tabular DL method TabM [6] has found ExcelFormer to be the second weakest baseline among 13 baselines compared, therefore we do not include it in the analysis.
>
>
>
> ### Summary of our response to the Weakness 2.
>
> Motivated by your concern about the breadth of the studied methods, we have added sections about data uncertainty robustness of two state-of-the-art approaches to Tabular DL: TabPFNv2/v2.5 and RealMLP. With these new additions, our analysis covers all the top non-foundational DL approaches from the TabArena [5] benchmark and two of the best foundational tabular DL models.

---

> > ### Author Response · Authors · 2025-11-25
> >
> > ## Weakness 3, Limited realism of real-world evaluations
> >
> > We higly value transferability of our results from academic settings to real-world situations. For this reason, in addition to the standard academic datasets, our paper uses TabReD benchmark [2], which was accepted as a Spotlight paper at ICLR 2025. The abbreviation TabReD stands for ''Tabular **Real** Data'', and the stated goal of the paper [2] was to produce a collection of datasets which are maximally close to a realistic tabular task. Since the investigation of realism of different datasets was outside the scope of our paper, we build on previous work and utilize TabReD as a source of realistic tabular datasets. In addition with our results described in our response to your Weakness 1 demonstrating applicability of our findings to different sources of data uncertainty, we believe our revised manuscript holds a high standard for the realism of studied settings.
> >
> > [1] Pattern Recognition and Machine Learning, Christopher Bishop, https://www.microsoft.com/en-us/research/wp-content/uploads/2006/01/Bishop-Pattern-Recognition-and-Machine-Learning-2006.pdf
> >
> > [2] TabReD: Analyzing Pitfalls and Filling the Gaps in Tabular Deep Learning Benchmarks, Rubachev & Kartashev et al., https://proceedings.iclr.cc/paper_files/paper/2025/hash/571799482291411607c54984153190b0-Abstract-Conference.html
> >
> > [3] On Finetuning Tabular Foundation Models, Rubachev et al., https://arxiv.org/abs/2506.08982
> >
> > [4] Better by Default: Strong Pre-Tuned MLPs and Boosted Trees on Tabular Data, Holzmüller et al., https://arxiv.org/pdf/2407.04491
> >
> > [5] TabArena: A Living Benchmark for Machine Learning on Tabular Data, Erickson et al., https://arxiv.org/abs/2506.16791
> >
> > [6] TabM: Advancing Tabular Deep Learning with Parameter-Efficient Ensembling, Gorishniy et al., https://arxiv.org/abs/2410.24210

---

### Official Review · Reviewer_1pkh · 2025-10-31

**Soundness:** 4
**Presentation:** 3
**Contribution:** 3
**Rating:** 4
**Confidence:** 3

**Summary:**

This paper studies the impact of label noise on different variants of DL models for predictive tabular tasks. To do so, the authors propose an experimental protocol to estimate label noise on real-world datasets (and validate this setup in synthetic tasks). As a result, they show that recent performance improvements by introducing better embeddings, ensembling, and retrieval-augmented methods to DL models are correlated with a better handling of noisy labels.

**Strengths:**

[Originality and Novelty] This paper presents a creative and well-executed empirical study on the impact of recent design choices. It offers a novel perspective and provides valuable insights into model behavior, contributing to the development of better models for predictive tabular tasks.

[Quality] Overall, I really enjoyed reading this paper. It is well written with useful visualizations, plots, and interesting observations. The experiments are well justified and clearly explained, and the findings are based on a solid empirical foundation supported by large-scale experimentation. The limitations section also provides an honest assessment, helping contextualize the findings' impact and significance.

**Weaknesses:**

[model choice] With the advent of many pre-trained foundation models for predictive tasks, an obvious question is how such models would perform in this experiment. Although the authors acknowledge the absence of such state-of-the-art models (and proposed to study them for future work), I would've hoped for a use case (where such models cannot be used) to provide a clear justification for why such models were not investigated. Additionally, since many experiments emphasize neighborhood-based methods, including a simple KNN baseline (e.g., in Figure 5) for comparison would be informative.

[robustness of results] The experiments are conducted on a large scale with HPO; however, the sensitivity of the results to hyperparameters, and especially model capacity, is not discussed. Reporting variability across repetitions or different synthetic dataset samples would also be valuable.

[generalization of results] While the experiments provide a strong signal, the empirical analysis is limited to a few hand-selected datasets and only ablates three design decisions, with notable overlap between dataset authors and model contributors. This observation raises questions about the generalizability of the findings to broader settings.

**Questions:**

My current rating reflects the paper in its present state (while I remain open to revising my opinion), considering the weaknesses discussed above and the questions listed below. The experimental framework is very useful and provides a valuable lens to neutrally study model behavior. However, the small range of datasets, models, and tasks studied limits the potential impact the paper could (and should) achieve. The authors are welcome to address any of the raised weaknesses, but I am particularly interested in their response to the following points:

1) [minor; clarification] The limitation section states that this work focuses on regression tasks (such as the datasets used in Section 5). However, the datasets in Sections 4.2 and 4.3 seem to be classification tasks. Is there something I'm missing?

2) [relation to RealMLP] It would be helpful if the authors could discuss how their findings relate to or might extend to RealMLP which, among other topics, also studies different encoding strategies; see https://dl.acm.org/doi/10.5555/3737916.3738753

---

> ### Author Response · Authors · 2025-11-25
>
> Thank you for the review.
>
> ## Weakness 1, model choice
>
> Most datasets in our analysis have sample sizes above 10000. In case of TabPFNv2, authors officially limit the use-cases for their model at 10000 samples. Nevertheless, we obtained the results for several datasets, and provide the corresponding uncertainty plots in Appendix P.
>
> Summarizing our findings, we show that Prior Fitted Networks (with TabPFNv2 and TabPFNv2.5 used as examples) outperform the MLP-baseline more also on the high-data uncertainty samples, similarly to the ModernNCA, TabM and numerical embeddings. While a complete mechanistic study of reasons behind this phenomenon remains to be too resourse-intensive, a recent paper [1], has shown a connection between the way PFN-like models and retrieval methods work, which could explain their shared advantage in niches of high data uncertainty. We consider this new result as a valuable addition to our work since it demonstrates that the concept of data uncertainty can be used to explain the TabPFN success as well.
>
> > including a simple KNN baseline (e.g., in Figure 5) for comparison would be informative.
>
> Unfortunately, because our analysis is sample-wise, it is not informative to compare the methods with very different performance, as large overall differences in quality can overwhelm any sample-wise dynamic. The KNN model in the raw feature space is significantly inferior to the MLP baseline on most of the datasets.
>
>
>
> ## Weakness 2, robustness of results
>
> In Appendix Q, we demonstrate how different hyperparameters like weight decay and dropout can have different degrees of infuence on robustness to data uncertainty. However, we do believe it is important to compare the different methods at their most optimal configuration.
>
> In future updates to the manuscript, we will also include
> 1. Scores across different samples of the Uncertain CIFAR-10 dataset, which demonstrate statistical meaningfulness of our result.
> 2. Uncertainty plots for models of different capacity, which demonstrate applicability of our findings on different scales.
>
> ## Weakness 3, generalization of results
>
> Motivated by the reviewer's suggestions, we have conducted the following additional experiments:
>
> 1. In Appendix R, we include results on the TabArena benchmark [2], which was filtered to include datasets with no less than 10k total samples. The results confirm the importance of the data uncertainty concept on the TabArena datasets as well.
>
> 2. In Appendix P, we include uncertainty plots for TabPFNv2 and TabPFNv2.5 models. These plots demonstrate that Foundational Tabular Models show robustness to high data uncertainty, which could be potentially explained by their relationship with retrieval methods([1]).
> 3. In Appendix P, we also provide uncertainty plots for RealMLP. These plots demonstrate that RealMLP shows robustness to high data uncertainty, which could be explained by both the advanced embedding scheme utilized in RealMLP as well as improvements to optimization which lead to more stable optimization in the precense of noise.
>
> We believe the additional experiments described above greatly extend upon the settings studied in the paper and increase the applicability of our analysis.

---

> > ### Author Response · Authors · 2025-11-25
> >
> > ## Question 1, minor; clarification
> >
> > ### Regarding the dataset in 4.3
> >
> > While data uncertainty is not a phenomenon exclusive to regression tasks, when estimating data uncertainty for classification tasks, finding a correct calibration technique provides an additional degree of freedom which could complicate analysis. Nevertheless, we believe increased data uncertainty in classification tasks also leads to increased performance in studied methods. The role of Uncertain CIFAR-10 dataset from 4.3 was twofold:
> >
> > 1. Show that our findings of studied methods' robustness to high data uncertainty hold even for classification tasks
> > 2. Show that the same findings hold on a dataset where data uncertainty comes not from added noise, but from removed relevant features.
> >
> > Motivated by other reviewers' concerns, we have expanded the study of classification problems in our manuscript:
> >
> > 1. In Appendix K, we provide uncertainty plots for popular classification datasets such as Adult and Higgs, as well as for our Uncertain CIFAR-10 dataset. We use the entropy of predicted probabilities as a measure of data uncertainty, following [3].
> > 2. In Appendix M, we provide the experiment on the version of CIFAR-10 with original features, but 50% symmetric label noise (in 50% of samples targets are changed to a random different label). We show that compared to MLP, results of the studied methods greatly improve on the noisy version of the dataset. The table with accuracy difference between MLP and studied methods is included below as ``Table 1``.
> >
> >
> > **Table 1.** Accuracy of different methods on original and noisy versions of the CIFAR-10 datasets. MLP-LRLR begins to outperform MLP when both are trained on noisy data, while ModernNCA starts showing almost the same performance as MLP, although it is about a percentage worse when trained on the noiseless version of the dataset. While TabM and Deep Ensemble show the same accuracy when trained on the noiseless version of CIFAR, TabM outperforms Deep Ensemble when noise is injected in the training data.
> >
> > | Method | CIFAR-10 | CIFAR-10 with 50% label noise|
> > | --- | --- | --- |
> > | MLP | 58.31+-0.27 | 46.28+-0.44|
> > | MLP-LRLR | 57.19+-0.47 | 48.20+-0.48|
> > | ModernNCA | 57.48+-0.22 | 46.20+-0.60|
> > | Deep Ensemble | 60.92+-0.22 |48.02+-0.27|
> > | TabM | 61.01+-0.25 | 48.63+-0.27|
> >
> > We believe additional experiments described above allow our work to demonstrate the applicability of our findings in the task of classification and additionally increase the number of datasets studied in the paper.
> >
> >
> > ### Regarding the dataset in 4.2
> >
> > The dataset described in 4.2, it is a regression dataset. While true unnoised target values can only take values of 0 and 1, the added noise is normal, and therefore noisy targets can have any real value.
> >
> > ## Question 2, [relation to RealMLP]
> >
> > We do believe that the advanced embedding strategy employed by the authors of RealMLP bears the same quality of robustness to data uncertainty as numerical embedding methods studied in our paper. Moreover, it has been observed that the presence of noise in the training data can complicate optimization [4, 5]. The steps the authors of RealMLP took to improve the process of gradient descent (changes in beta_2 in Adam, learning rate schedule, etc.) could improve performance in the situations of high data uncertainty even further.
> >
> > In Appendix P of the revised manuscript we provide uncertainty plots for RealMLP that demonstrate its robustness to high data uncertainty.
> >
> > ## Summary of the new results
> >
> > In our rebuttal we provided new experimental results, extending:
> > 1. The set of datasets for which we demonstrate the studied effect (by TabArena, several classification datasets, and a new synthetic CIFAR-10 with 50% symmetric noise)
> > 2. The set of methods for which we demonstrate the studied effect (TabPFNv2, TabPFNv2.5, RealMLP)
> > 3. The set of tasks for which we demonstrate the studied effect (added classification)
> >
> > In our answer to weakness 2, we also provide a study on the relationship between hyperparameters and robustness to data uncertainty.
> >
> > In the future versions of the manuscript, we will include:
> >
> > 1. Scores of our methods on different generations of Uncertain CIFAR-10 dataset (in each version different 50 features are left in the data)
> > 2. Uncertainty plots for models with three different levels of model capacity.
> >
> > [1] On Finetuning Tabular Foundation Models, Rubachev et al., https://arxiv.org/abs/2506.08982
> >
> > [2] TabArena: A Living Benchmark for Machine Learning on Tabular Data, Erickson et al., https://arxiv.org/abs/2506.16791
> >
> > [3] Uncertainty in Gradient Boosting via Ensembles, Malinin & Ustimenko & Prokhorenkova, https://arxiv.org/pdf/2006.10562
> >
> > [4] On the Role of Label Noise in the Feature Learning Process, Han & Huang & Zhou et al., https://arxiv.org/pdf/2505.18909
> >
> > [5] Learning from Noisy Labels with Deep Neural
> > Networks: A Survey, Song et al., https://arxiv.org/pdf/2007.08199

---

> > > ### Author Response · Authors · 2025-12-04
> > >
> > > We've now updated the manuscript with results we've promised earlier in the rebuttal. Appendix S comparing models of different capacity and Appendix T with variablility across diffirent Uncertain CIFAR dataset generation seeds.

---

### Author Response · Authors · 2025-12-03

Dear Area Chair, reviewers and readers.

In this paper, we conduct an empirical study that demonstrates how important the concept of data uncertainty is for Tabular Data. A big part of our analysis relies on analysing performance of different methods on different data samples in a tabular dataset through the prism of their data uncertainty. We first note that GBDTs outperform a simple MLP neural network especially in regions of the dataspace where data uncertainty is the highest. We also observe the same effect for several recent state-of-the-art Tabular DL approaches. We then provide validation of our data uncertainty estimation procedure and demonstrate the relationship between data uncertainty and performance of Tabular DL methods on large number of datasets. Importantly, we show that the concept of data uncertainty provides a unified explanation scheme of the recent methods' empirical success.

Four reviewers ***unanimously agree*** with the importance of our work as a unified explanation for the success of several state-of-the-art methods in Tabular DL. To quote the reviews specifically:

Reviewer 1pkh:

> It offers a novel perspective and provides valuable insights into model behavior, contributing to the development of better models for predictive tabular tasks.

> The experimental framework is very useful and provides a valuable lens to neutrally study model behavior.

>The limitations section also provides an honest assessment, helping contextualize the findings' impact and significance.


Reviewer ewbA:

> The study provides a novel perspective linking model design and data uncertainty in tabular learning.

> The paper provides a clear and well-motivated analysis of why certain design choices in tabular deep learning [...] might succeed. Framing these differences through the lens of data uncertainty offers a fresh and conceptually coherent perspective.


Reviewer ixYt:

> Overall, this paper provides a set of results that help explain the success of recent tabular DL models

> I think this is a very important contribution to the field, given that this same discussion has been brought up multiple times (WhyTrees, Tabzilla) but not fully explored.

> The findings from this paper seems to pinpoint a solid factor in the success of GBDT models, and also provide a path forward for tabular DL models to close the gap.

Reviewer nGFb:

> The paper proposes a unifying, sample-wise lens—data (aleatoric) uncertainty—to explain why several recent choices in tabular DL tend to help.

> Clear, unifying viewpoint that organizes disparate tabular DL results under one explanatory factor (data uncertainty). The uncertainty-slice analysis feels insightful and actionable.


Most of the reviewers have also found that the experimental basis for our claims is solid:

Reviewer 1pkh:

> The experiments are well justified and clearly explained, and the findings are based on a solid empirical foundation supported by large-scale experimentation.

> Overall, I really enjoyed reading this paper. It is well written with useful visualizations, plots, and interesting observations.

Reviewer ixYt:

> The experiments are well designed and the results are clearly presented.

Reviewer nGFb:

> Careful sanity checks: synthetic datasets with known uncertainty; robustness of findings to different uncertainty estimators (Appendix I).

> Mechanistic probes beyond benchmarks (e.g., neighbor-target consistency, gradient clean/noisy decomposition for TabM).

> Breadth: results cover several public tabular benchmarks; includes a pruned “Uncertain CIFAR-10” to stress uncertainty effects across settings.

Given the unanimous confirmation of the novelty and practical importance of our uncertainty-centric perspective in Tabular DL, we believe our study provides a valuable contribution for the field, especially after providing the additional results requested by the reviewers.

The reviewers 1pkh and ewbA, despite noting the importance of our main contribution, have concerns on the breadth of the setups we investigate. Reviewer 1pkh requested more datasets, as well as analysis for such state-of-the-art approaches as RealMLP and TabPFN, while reviewer ewbA requested analysis with more sources of data uncertainty, as well as the analysis for other state-of-the-art methods.

---

> ### Author Response · Authors · 2025-12-03
>
> In our responses to reviewers, we provided results that, according to our belief, address all of their concerns. More specifically, below we provide a list of the experiments added in our rebuttals:
>
> ## Reviewer 1pkh
>
> 1. In Appendix P, we added experiments for TabPFNv2, TabPFNv2.5, and RealMLP. This directly addresses reviewer's request for analysing more models with different authors.
> 2. In Appendix Q, we demonstrate that there is no clear influence of model's hyperparameters like weight decay and dropout on a method's robustness to data uncertainty. In our response, we also restate our belief in comparing methods' behavior in their best hyperparameter configuration, and note that we used analogous hyperparameter tuning procedure across the tested models.
> 3. In Appendix S we've added uncertainty plots for models with three different levels of model capacity.
> 4. In Appendix R, we provide results on the TabArena benchmark, which we filtered to include only datasets with medium to large sizes. This directly answers the request made by the reviewer to increase benchmark diversity. We do not analyze datasets with total sizes less than 10,000 samples, because we study small subregions of the test subset of the data, which is hard to do with extremely small test sizes. Additionaly, we did not verify our data uncertainty estimation procedure in the small dataset niche. It may be harder to achieve a robust data uncertainty estimation procedure on small datasets.
> 5. In Appendix K we provide uncertainty plots for classification datasets such as Adult, Higgs, and our synthetic dataset Uncertain CIFAR-10. This helps answer reviewer's question about the studied effect in conditions of classification.
> 6. In Appendix M, we conduct experiments on the new synthetic dataset which was built by adding 50% label noise to CIFAR-10. We show increases in accuracy relative to the baseline MLP for all studied methods on yet another classification dataset, which also re-confirms applicability of our findings.
> 7. In appendix T we've added scores across four more different samples of the Uncertain CIFAR-10 dataset, which demonstrate statistical meaningfulness of our result.
>
> ## Reviewer ewbA
>
> 1. In our response to weakness 1, we reproduce our findings when using different sources of data uncertainty:
>     - In Appendix L, we conduct experiment on a synthetic dataset with assymetric noise. The dataset is akin to the one introduced in 4.1, but with skew-normal noise distribution used instead of just normal.
>     - In Appendix M, we conduct experiments on the version of CIFAR-10 dataset which 50% label noise, and demonstrate that the studied methods show improved difference in performance to the MLP with added label noise.
>     - In Appendix K, we show that on the task of classification, when data uncertainty is estimated as the entropy of the predicted probabilities and noise does not follow gaussian distribution, our findings still apply. Moreover, in the case of Uncertain CIFAR-10, this also demonstrates robustness to data uncertainty which arises from missing features.
>
> 2. In our response to weakness 2, we provide results for RealMLP and TabPFNv2, which directly answers the reviewers concern about our analysis not covering some of the state-of-the-art methods.
>
> 3. In our response to weakness 3, we disagree with the claim about lack of realism in the choice of our benchmark, since our paper includes experiments on TabReD, which claims to be the best representation of realistic tabular data. Nevertheless, we have extended our experiments to TabArena benchmark, with results included in our response to the reviewer 1pkh.

---

> > ### Author Response · Authors · 2025-12-03
> >
> > ## Reviewer ixYt
> >
> > In our response to the reviewer, we include the experimental results on binary classification datasets, which directly answer the first weakness noted by the reviewer:
> >
> > 1. In Appendix K we provide uncertainty plots for datasets such as Adult, Higgs, and our synthetic dataset Uncertain CIFAR-10.
> > 2. In Appendix M, we conduct experiments on the new synthetic dataset which was built by adding 50% label noise to CIFAR-10. We show increases in accuracy relative to the baseline MLP for all studied methods on yet another classification dataset.
> >
> > ## Reviewer nGFb
> >
> > 1. In Appendix O, we study sample-wise changes in performance for ModernNCA, TabM and MLP for the diamond dataset, to which we add uniform normal noise. This answers the reviewer's second question.
> > 2. In Appendix K we provide uncertainty plots for datasets such as Adult, Higgs, and our synthetic dataset Uncertain CIFAR-10. This directly answers reviewer's weakness 3.
> > 3. In Appendix N, we show differences in performance sorted by epistemic uncertainty instead of the aleatoric uncertainty on the maps-routing dataset. This shows that on a dataset where aleatoric and epistemic uncertainty are not strongly correlated, the studied effect occurs only for the aleatoric uncertainty. This experiment addresses one part of weakness 4.
> > 4. In Appendix Q, we provide uncertainty plots for weight decay and dropout. This experiment addresses other part of weakness 4.
> >
> >
> > We also address some of the other issues / questions without conducting new experiments in our full rebuttal.

---

### Meta-Review · Area_Chair_1zYN · 2026-01-07

**Summary:**

**Summary:** This paper focuses on tabular deep learning. It identifies that successful tabular DL models outperform naive ones (e.g., MLP) mainly in high data uncertainty regimes. It then further provides explanations and introduces embedding learning as a further improvement.

**Strength (by reviewers):** creative, well-executed, unifying, and well-motivated empirical study & analysis; novel, potentially impactful insights; well-written paper; careful sanity check and mechanistic probes

**Weakness (by reviewers):** limited model choices and datasets (unclear generalizability); missing strong modern models; narrow treatment of data uncertainty; the observations is dependent on specific uncertainty modeling; unvalidated uncertainty estimates on real data; only focus on regression (the AC thinks this is fine); unclear result robustness; missing technical details; slightly inconsistent content (starting with GBDT but then focusing on advanced DL methods)

**Decision:** The paper received an average score of 4.5 (2, 4, 6, 6). The authors provided extensive rebuttals; however, none of the reviewers were involved in the discussions. The AC read the paper, reviews, and the rebuttals. Many of the concerns/weaknesses have been addressed (at least partially). The paper's score is likely to increase to 5 (3, 5, 6, 6)---thus, a borderline. The AC agrees with the strengths pointed out by the reviewers: the observations and analyses are insightful and potentially impactful. However, the AC also agrees with some of the limitations pointed out by reviewers (though the AC thinks it is fine to focus the study on regression only).

While reading the paper, the AC finds some further concerns.
- While the **observation** that modern DL methods outperform naive MLP mainly in high data uncertainty regimes is interesting, the current paper lacks *further, more sound analysis on why MLP performs worse in this regime and how this regime can be improved.* (This would serve as future design principles and as a unifying bridge for understanding modern DL methods, such as ModernNCA, TabM, and MLP-LRLR.) Please note that, according to the definition of epistemic and data (or aleatoric) uncertainty (lines 112 - 116), data uncertainty is independent of the specific model used. In other words, ModernNCA, TabM, and MLP-LRLR wouldn't resolve data uncertainty. (Numerical feature embeddings are unlikely to address it, as they won't introduce additional mutual information between X and Y.) So why do they outperform MLP? The AC hypothesizes that the improvement may indeed come from their better handling of epistemic uncertainty. Specifically, for data uncertainty, the best a model can do is to predict the conditional mean (Bayes Optimal in the MSE metric). However, for samples with large uncertainty, robustly estimating their means needs more data. This is likely where ModernNCA, TabM, and MLP-LRLR show benefit.
- Claims in sections 5 & 6 need further justification/clarification. For example, lines 348-358 read more like observations and hypotheses, not sound explanations. Section 5.1 seems to suggest that MLP overly smooths the prediction, but Section 6 suggests that ModernNCA outperforms MLP because it leads to less overfitting. These two claims/explanations seem to contradict. Finally, *learning embeddings to pull data with similar targets together* is a fundamental technique to improve traditional ML (tabular data), e.g., metric learning, supervised dimensionality reduction. It is indeed one design choice in ModernNCA. Its improvement is somewhat expected; whether it really addresses data uncertainty is unclear.

With these additional concerns, the AC recommends rejection. The AC does see that this research has the potential to be of high impact. However, the current manuscript does need a major revision. Publishing it in its current form would do a disservice to the research and the community. The AC suggests that the authors focus on regression, as it is easier to analyze epistemic and aleatoric uncertainty in regression. Specifically, the AC suggests that the authors consider a more formal analysis based on the well-known bias-variance-data-uncertainty decomposition and Bayes-optimal predictors.

**Reviewer Concerns:**

**Reviewer 1pkh.**
- **Addressed:** including some more SoTA (e.g., pre-trained) models; unclear result robustness (to a certain extent); more datasets
- **Remained:** k-NN (while AC thinks ModernNCA is considered a k-NN-based method); classification tasks (some results are presented, but the overall trend does not really align with regression tasks)

**Reviewer ewbA.**
- **Addressed:** narrow treatment of data uncertainty (to a certain extent, but may not be fully addressed); including strong modern models; limited real data (to a certain extent)
- **Remained:** see above

**Reviewer ixYt.**
- **Addressed:** clarification questions and examples
- **Remained:** classification tasks (some results are presented, but the overall trend does not really align with regression tasks); the observations are dependent on specific uncertainty modeling

**Reviewer nGFb.**
- **Addressed:** unvalidated uncertainty estimates on real data (partially); clarification on experiment design choices (partially); additional experiments and comparison; classification data
- **Remained:** see above

**Reviewer Scores:**

**Reviewer 1pkh (4 to 5):** concerns were partially addressed; the reviewer may keep the score as 4 or increase it to 6 (thus 5).

**Reviewer ewbA (2 to 3):** concerns were partially addressed; the reviewer may keep the score as 2 or increase it to 4 (thus 3).

**Reviewer ixYt (6 to 6):** concerns were partially addressed, but the AC does not think the rebuttal would boost the score to 8.

**Reviewer nGFb (6 to 6):**  concerns were partially addressed, but the AC does not think the rebuttal would boost the score to 8.

Please note that, to the AC's humble opinion, addressing review concerns does not mean the score will definitely increase. Often, reviewers keep their original scores when their concerns are addressed, especially if the original scores were already positive.

---

### Decision · Program_Chairs · 2026-01-26

Reject